# Spatial contexts with reliable neural representations support reinstatement of subsequently placed objects

Rolando Masís-Obando [1,2] ✉, Kenneth A. Norman [1,3] &
Christopher Baldassano [4]

What are the neural properties that make spatial contexts effective scaffolds for storing and accessing memories? Here we hypothesized that spatial locations with stable and distinctive (that is, reliable) neural representations would best support memory for new experiences. To test this, participants learned the layout of a custom-built 23-room virtual reality 'memory palace' that they explored using a head-mounted display. The next day, participants underwent whole-brain fMRI while watching videos of the rooms, allowing us to measure the reliability of the neural activity pattern associated with each room. Participants then returned to virtual reality to encode 23 objects placed in each of the 23 rooms and later recalled the rooms and objects during fMRI. We found that our room reliability measure (computed before encoding) predicted object reinstatement during recall across cortex; this was driven not only by group-level reliability across participants but also by idiosyncratic reliability within participants. Moreover, this effect did not arise through enhanced retrieval of reliable rooms during recall, because the relationship between reliability and object reinstatement remained significant when controlling for room reinstatement during retrieval; this suggests that, instead, room reliability promotes improved binding of rooms to objects at encoding. Together, these results showcase how the quality of the neural representation of a spatial context can be quantified and used to 'audit' its utility as a memory scaffold for future experiences.

Many of our memories are intrinsically tied to the locations where they occurred. Thinking about (or actually revisiting) places from our past can immediately bring to mind the meaningful events that occurred there. In this way, our spatial memories can serve as a map not only of physical spaces but also of our remembered experiences in those spaces. In what ways can a spatial context (that is, the location in which an experience takes place) serve as a scaffold for storing and accessing the details of past episodes? Are there spatial contexts that are more

or less effective for attaching event memories, and can we neurally measure the usefulness of a location as a memory cue even before an event has occurred?

Decades of research have found that the representation and retrieval of episodic memories is profoundly tied to spatial location. Prior behavioural research on the context-dependent memory effect suggests that items learned in a particular physical context can be better remembered when the retrieval context matches the encoding context[1,2],

[1]Princeton Neuroscience Institute, Princeton University, Princeton, NJ, USA. [2]Psychological and Brain Sciences, Johns Hopkins University, Baltimore, MD, USA. [3]Department of Psychology, Princeton University, Princeton, NJ, USA. [4]Department of Psychology, Columbia University, New York, NY, USA. ✉e-mail: rmasis@princeton.edu

**Fig. 1 | Experimental paradigm.** Participants played a set of foraging games to learn the layout of the 23-room VR environment (photograph of a laboratory member demonstrating the VR set-up used with permission). At the end of each foraging game, to test learning of environment, participants drew a bird's-eye-view map of the environment (see Supplementary Fig. 1 for more examples). Twenty-four hours later, participants were shown room videos in the scanner, with each room presented twice. Participants then re-entered immersive VR and were given 15 min to learn the identities and locations of 23 new objects that had been added to the environment, one per room. Finally, participants returned to the scanner and recalled the items they had seen in a free recall task, a guided recall task (in which they recalled items along specific five-room paths) and a room video task (in which they recalled the item for each presented room). They were also presented with videos of each object and attempted to recall the room in which each object appeared.

even for contexts that are experienced only through virtual reality (VR)[3] or that are mentally reinstated rather than physically re-experienced[4]. Recent behavioural work has also suggested a privileged role for spatial contexts as cues for memory retrieval. For example, spatial context cues: (1) enhance episodic recall when compared with temporal, thematic (for example, romantic experience), person or object cues for imagined or real autobiographical memories[5–8]; (2) are spontaneously generated even when not cued by experimenters[6,9], sometimes leading to quicker access to episodic information[6,9] (but see ref. 10); (3) are associated with richer episodic memory when highly familiar to participants[6,9,11–13]; and (4) are associated with preserving long-term recollection of initially low

detail memories for both young and older adults[14]. This behavioural work is complemented by neuroimaging studies of autobiographical memory showing that spatial contexts have a strong influence on the neural representations of remembered or imagined autobiographical events[9,15,16], among others; for a review, see ref. 17. The networks associated with spatial contexts are maintained during multiple phases of memory retrieval, possibly acting as a scaffold for accessing additional event details[18]. For example, spatial contexts can be reinstated before or concurrently with the retrieval of an item or episode[19,20].

Beyond retrieval, prior theoretical work on episodic memory suggests that—at encoding—features of an ongoing experience are bound

to the context in which they occur[21–23], allowing spatial contexts to serve as structured 'containers' that organize and support the integration of new experiences[24,25]. Consistent with this view, explicitly binding objects to their spatial context during encoding enhances subsequent memory for those objects[26]. In another recent study[3], participants encountered words within two distinct spatial contexts (each associated with a separate schema) and judged each word's relevance to its context without knowing there would be a later memory test. If reinstating the context at retrieval were sufficient to boost memory, all words should have benefitted equally. Instead, only context-relevant words showed a memory advantage, suggesting that these items were more effectively bound to the spatial context during encoding.

Despite the centrality of spatial context in memories, it is unknown whether (1) some specific spatial locations are more effective memory cues than others and, if so, (2) whether this is related to properties of their neural representation. In general, two requirements for robust representation of a memory are thought to be stability over time (allowing for faithful reactivation of the features of the original experience) and distinctiveness (to prevent interference with other similar memories[27,28]). We hypothesized that these two properties would also be important specifically for building an effective spatial context scaffold—that is, that spatial locations with more stable and distinctive representations would support better encoding of new information encountered in these locations and allow easier access to this information at retrieval. This implies that having a stable and distinctive neural representation for a location before associating an object to that location will be predictive of subsequent reinstatement for that object representation.

Our primary mechanistic hypothesis for why this would occur was that reliable room representations facilitate the binding of room to object information at encoding (for example, the sturdier a wall is, the easier it is to hang a painting on it). However, facilitated binding at encoding is not the only way that having a stable and distinctive room representation could facilitate subsequent object reinstatement; an alternative possibility is that having a stable and distinctive room representation has no effect on room–object binding at encoding and that instead it boosts object recall indirectly by boosting the degree to which the room representation is reinstated at test, which—in turn—boosts reinstatement of associated object information (for example, the brighter the light in a dark room, the easier it is to see what is inside). We will present the results of analyses that control for this alternative possibility.

To test whether reliable spatial contexts scaffold subsequent memory, we custom-built a VR 'memory palace' environment of 23 perceptually distinct rooms each with distinct soundtracks, interiors and room-congruent objects, which participants explored using a head-mounted VR display (Fig. 1). After participants learned the layout of the virtual environment, we used functional magnetic resonance imaging (fMRI) to compute a neural room reliability score for each of the 23 rooms (Fig. 2). This score reflected both the stability and distinctiveness of neural representations, measuring the degree to which repeated presentations of a room evoked patterns that were more similar to each other than to patterns evoked by other rooms. Participants then returned to the VR environment, where they observed (and were asked to memorize) a new salient object that had now been placed into each room. Finally, they performed recall tasks for these items in the fMRI scanner (Fig. 3). Overall, our results confirmed our hypothesis: room reliability, measured before any room–object pairing occurred, predicted the degree of object reinstatement during verbal recall, showing that it is possible to neurally diagnose whether a room will serve as an effective memory scaffold, before objects are placed in the room.

## Results
### Overview
How effective are spatial memory representations as containers for subsequently bound objects? We sought to answer this question by using the reliability of a prelearning room representation to predict the degree of reinstatement evidence for recalled objects during self-paced verbal recall. To do this, we needed to quantify (1) the reliability of a room representation and (2) the reinstatement of object information during recall. We defined room reliability as the similarity of a room representation to itself (that is, stability) minus its average similarity to every other room (that is, distinctiveness); importantly, this was measured before any room–object associations had been formed (that is, in the prelearning phase; Fig. 2). Our strategy for quantifying object reinstatement during recall was as follows: We first identified a network of regions involved in the retrieval of objects (the retrieved object classifier network; ROCN) during a cued-recall task in which participants watched videos of room interiors and were asked to recall the objects that had been randomly assigned to those rooms in VR (Fig. 4). We then measured the average classifier evidence for object reinstatement within this network during self-paced verbal recalls, in which participants were instructed to verbally describe with as much detail as possible the rooms and the randomly placed objects in them (Fig. 5a). Afterwards, to determine how well the reliability of a prelearning room representation predicted object reinstatement, we correlated prelearning room reliability scores with object classifier evidence within the ROCN during self-paced recall trials (Figs. 5b and 6). We identified a set of regions whose prelearning room reliability predicted object reinstatement during verbal recall, including the precuneus, posterior parietal cortex, and prefrontal cortex—specifically, the superior frontal gyrus. Importantly, using a model comparison analysis, we also found that some of these regions provided a participant-specific predictive benefit, including the posterior parietal cortex, posterior ventral temporal cortex and superior frontal gyrus (Fig. 6b). Lastly, to identify whether room reliability supported object reinstatement indirectly by promoting room reinstatement at recall, we conducted a partial correlation analysis controlling for room reinstatement. Even after statistically controlling for room reinstatement, the relationship between room reliability and ROCN object reinstatement remained significant (Fig. 6c). Furthermore, no areas showed a significant decrease in the size of this relationship when we controlled for room reinstatement (see 'Partial correlation analysis controlling for room reinstatement' section in the Methods).

### Room reliability
To identify brain regions with reliable room representations for every participant, we compared the similarity of a room's representation across runs to its similarity with representations of other rooms (Fig. 2a). We ran this analysis on searchlights and hippocampal regions of interest (ROIs; full hippocampus, anterior hippocampus and posterior hippocampus). We found significant room reliability across most of the cortex. Unsurprisingly, given the audiovisual nature of the room videos, we found high reliability scores in the auditory and visual cortex, as well as in the precuneus and posterior hippocampus (Fig. 2c).

Are there particular room properties, such as size, complexity or connectedness, that contribute to the reliability of room representations? To identify which room features contribute to room reliability, we ran a searchlight analysis where, within each searchlight, we ran a multiple regression predicting room reliability based on six different room features; we generally found that, in default mode network regions, the most reliable rooms tended to be those that were small, had many corners and had an opening with a view to the outside (Supplementary Fig. 2).

### Behavioural recall
On the second day, participants performed two types of self-paced verbal recall task. During the guided recalls (11 runs), participants were presented with the names of 5 rooms that followed a path within the virtual palace and were asked to freely recall details of the rooms and the randomly added objects. During the free recalls, participants were

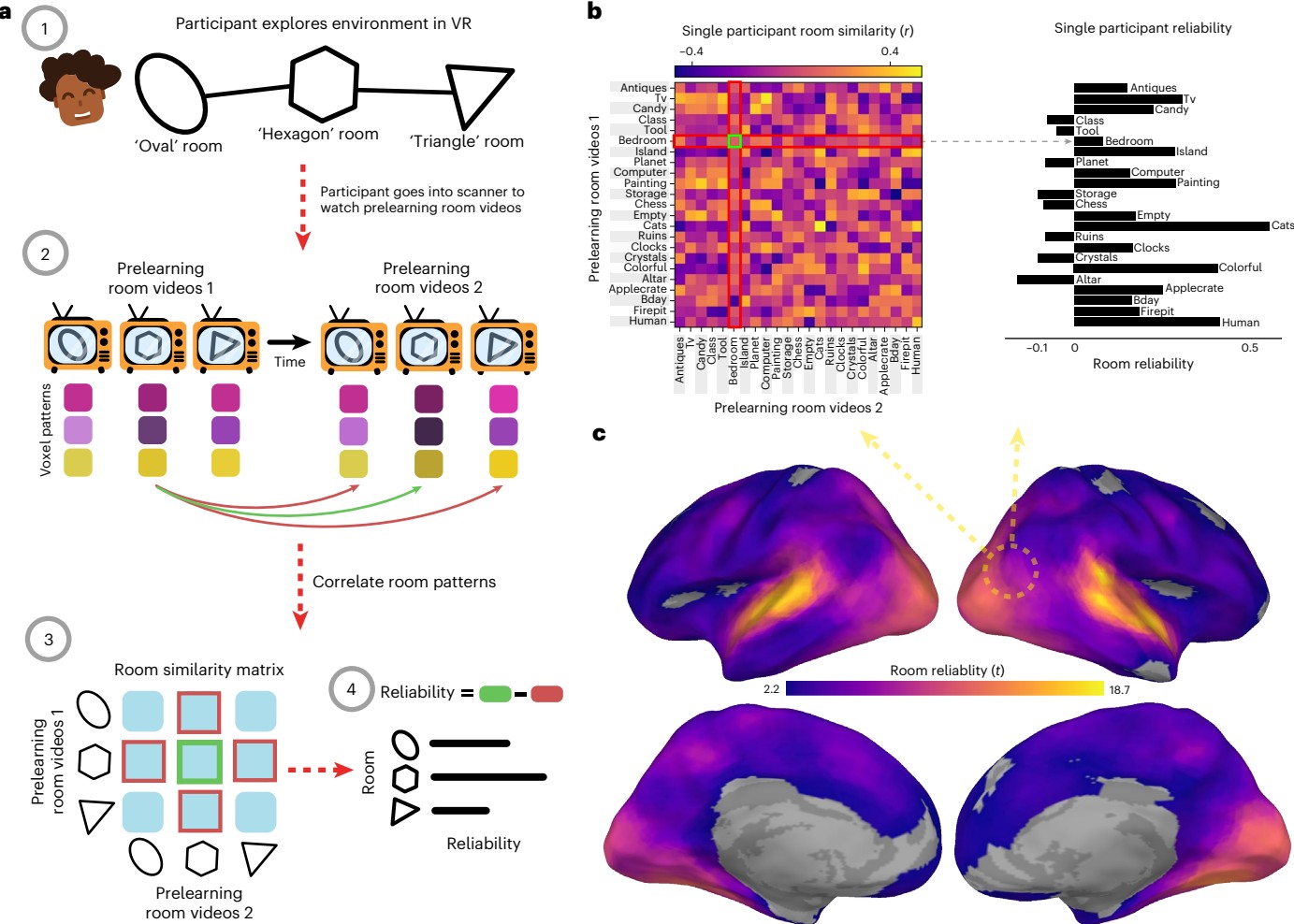

**Fig. 2 | Room reliability. a**, Illustration of room reliability methodology. (1) Participants first explored the memory palace in immersive VR and learned its spatial layout by playing a foraging game. (2) In the prelearning scanning session, before learning room–object pairings, participants watched and listened to videos of each room twice (prelearning room videos 1 and 2). Each room representation was correlated across runs with every other room, in a searchlight analysis. (3) Room reliability was computed by taking the difference between the similarity of a room pattern to itself (green) and the average similarity of the room with every other room (red). (4) Room reliability was computed for every room, leaving a room reliability score for each of the 23 rooms. The entire procedure outlined in **a** was computed for every participant such that, for every searchlight, there were 23 room reliabilities for each of the 25 participants. **b**, An example room pattern similarity matrix for one participant, in the searchlight denoted with a dotted circle. This matrix was used to extract room reliability scores as described in **a**, such that, for each room (row), the average room similarity to other rooms (red) was subtracted from the room similarity to itself (green). **c**, Room reliability across the brain. Coloured vertices on the surface indicate regions in which room reliabilities were significantly above zero at the group level ($q < 0.05$), with brighter colours indicating greater reliability.

presented with a blank screen and were simply asked to freely recall, in as much detail as possible, the rooms and the added objects. For the guided recalls, we computed accuracy by counting whether a participant recalled the randomly placed objects in that path regardless of whether they were correctly recalled in order of the path or with the correct room–object pairing. In other words, an object was marked as correctly recalled (out of 5) if it was recalled at any point during the trial. Similarly, for the free recalls, regardless of when an object was recalled, we marked an object as correctly recalled (out of 23) if it was recalled at any point during the free recall. Across both recall types, participants' recalls were at ceiling, with 92% and 80% of participants scoring higher than 90% recall accuracy for guided and free recalls, respectively (Fig. 3d). We also found that, in both guided and free recalls, participants spent less time speaking about the 'empty room' than the across-participant average (Supplementary Fig. 3)—probably because the room was empty (other than the randomly placed object) and there was less to recall. We also measured the proportion of contiguous room transitions during free recall. Across participants, spatially adjacent

rooms were recalled more often than expected by chance ($t(24) = 14.19$, $P < 0.001$), suggesting an unprompted bias towards contiguous mental traversal (Supplementary Fig. 3f).

## ROCN

To measure evidence of object reinstatement during self-paced (guided and free) verbal recall, we first needed to identify a network of regions that represent information about specific objects that were retrieved from memory; to select these regions in a non-circular fashion, we defined these regions using data from room–video object recall trials (Fig. 4a). In these trials, participants viewed videos of all rooms and verbally recalled the object that had been assigned to each room as it was presented (Fig. 4a). We used a leave-one-participant-out cross-validation procedure, whereby we made a neural template for each object (using data from a separate phase of the study in which participants viewed object videos) based on object videos from $N-1$ participants, and then we used these templates to classify the (not-visibly-present) objects being recalled during room viewing in

the held-out participant (Fig. 4a). We opted for this across-participant approach (rather than classifying within-participants) because objects and rooms are confounded within participants, so room information could 'leak' into training of a within-participant object classifier; this confound does not exist if training and testing are done across participants, each of whom has their own random set of room–object pairings. In other words, the left-out participant's object templates were never used to classify their own object recall during room videos. We used this procedure to identify the top 50 best object classifier searchlights (~3% of all searchlights) to make our ROCN (Fig. 4b), which we used as a mask (Fig. 4d) when measuring object reinstatement evidence during the guided and free recall tasks. We found that the top classifier searchlights were spread throughout cortex and included regions in the anterior temporal cortex, frontal gyrus, posterior temporal cortex, posterior medial cortex and superior parietal cortex, among others (Fig. 4c,d). We also conducted additional analyses to extract two other networks: For one, we classified object patterns while participants watched videos of objects (rather than retrieving object memories) to extract the perceived object classifier network (POCN), which was entirely, and unsurprisingly, due to the visual task, concentrated in early visual cortex (Supplementary Fig. 4). For the other, we classified room patterns while participants watched videos of objects (analogous to ROCN, which classified object memories during room videos) to extract the retrieved room classifier network (RRCN), which was widely distributed and included the precuneus, medial prefrontal cortex, anterior temporal cortex and visual cortex (Supplementary Fig. 5).

## Relationship of room reliability and ROCN object reinstatement evidence

Does room reliability predict future object reinstatement during free and guided recalls? Using the object classifier and ROCN searchlights from the previous analysis, we measured the degree of object reinstatement as each participant performed verbal recalls (Fig. 5a). Note that using neural object reinstatement provided a more sensitive index of successful retrieval than behavioural recall accuracy, as almost all participants were near-ceiling in their retrieval accuracy as described above. Specifically, at each searchlight, we correlated each participant's room reliability with their own composite ROCN object reinstatement score (Fig. 6a; see Supplementary Fig. 8 for an example searchlight). We then averaged these correlations across participants to obtain a searchlight map that we then statistically averaged across recall task types (that is, guided and free recalls) to get a composite map that indicated regions where room reliability in those regions correlated with subsequent object reinstatement (throughout the ROCN network; Fig. 6a). Notable positive relationships were observed throughout the parietal cortex, prefrontal cortex, superior frontal gyrus, insula and precuneus. We also found notable negative relationships in the right parahippocampal cortex, parts of the motor system, auditory cortex and ventral visual regions. Importantly, when looking at this

relationship separately for guided and free recalls (before generating our composite map), the regions revealed were highly similar, providing an internal replication of this relationship across two categorically different recall task types (Supplementary Fig. 7).

Lastly, to determine whether room reliability's relationship with object reinstatement was driven by room reinstatement, we ran a partial correlation analysis where we regressed room reinstatement scores in RRCN from both ROCN object reinstatement and prelearning room reliability, and then correlated the residuals. After controlling for room reinstatement at retrieval, the relationship between room reliability and ROCN object reinstatement evidence remained significant (Fig. 6c). The pattern of results across the brain shown in Fig. 6c (when we controlled for room reinstatement) was almost identical to the pattern of results shown in Fig. 6a (when we did not control for room reinstatement), and there were not any areas where the effect significantly differed between the two maps. Taken together, these results indicate that fluctuations in room reinstatement during retrieval were not responsible for the effects shown in Fig. 6a. For completeness, we also did this for POCN object reinstatement; similarly to what we found for the ROCN, after controlling for room reinstatement at recall, the relationship between room reliability and POCN object reinstatement remained significant, and there were no areas where this relationship significantly decreased when we controlled for room reinstatement (Supplementary Fig. 6c).

To what extent do the effects in Fig. 6a reflect group-level differences across rooms (whereby some rooms have both high reliability and high item reinstatement in all participants) versus participant-specific differences in which rooms are most reliable in their individual mental maps? To answer this question, we compared the coefficient of determination ($R^2$) between (1) our original participant-specific model, where each participant's object classifier evidence was predicted using their own room reliability values, and (2) the average $R^2$ of $N-1$ models where—in each model—the left-out participant's object classifier evidence was predicted using a different participant's room reliability values (that is, one model for each of the $N-1$ other participants). We then took the regions where there was a positive and statistically significant participant-specific effect (that is, better prediction with the original model) and intersected them with the correlational analysis performed in Fig. 6a. This process revealed a participant-specific benefit of room reliability in the posterior parietal cortex (near the angular gyrus), insula and superior frontal gyrus (Fig. 6b). Interestingly, there was also a participant-specific effect where room reliability in a small section of right parahippocampal cortex was negatively associated with ROCN reinstatement evidence.

In a similar fashion to how we related room reliability with object evidence within the ROCN, we ran a supplementary analysis in which we quantified object reinstatement within the POCN; largely composed of visual regions) during verbal recall (Supplementary Fig. 6). Across participants, we found generally similar results to the ROCN results,

**Fig. 3 | Behavioural recall scoring.** On the second day after learning room–object associations in VR, participants went back into the scanner where they performed a free recall and 11 guided recall tasks. In the free recall task, participants were asked to recall and describe with as much detail as possible the rooms and the objects paired to them. By contrast, during the guided recall tasks, participants were presented with five contiguously connected rooms and asked to describe the rooms and the objects in them. **a**, Example guided recall transcription. Participant recalls were transcribed manually for the onset and offset timestamps of when rooms and objects were recalled. **b**, Example transcribed recall event matrix. Timestamps of the onsets and offsets of participant recalls were then interpolated from seconds into TRs and organized as event matrices that could then be used to index BOLD recall timeseries. Green and yellow bars indicate room and object recalls, respectively. Object recall timepoints were used to calculate object evidence scores in neural analyses (for example, the timepoints where participants talked about the dartboard object ('Darts')

were used to measure neural object evidence for recall of the Darts). **c**, Guided recall task, pairings, objects recalled and accuracy calculation. First column: this participant was presented with a five-room path and asked to sequentially describe the rooms and the objects in them. Second column: calculation of behavioural accuracy for guided recalls. Participants were scored based on whether they recalled the objects that were paired to the rooms in the presented path. Although this participant recalled four objects, only three were associated with the corresponding cued five-room path. For both guided and free recalls, points were awarded based on whether participants recalled the relevant objects at any point in time during the recall period, regardless of the order in which the objects were recalled or whether they were recalled in association with the correct room. **d**, Guided recall (GR) and free recall (FR) accuracy distributions. With these scoring schemes, participants were able to recall objects with high accuracy. Participants' recalls were at ceiling, with 92% and 80% of participants scoring higher than 90% recall accuracy for guided and free recalls, respectively.

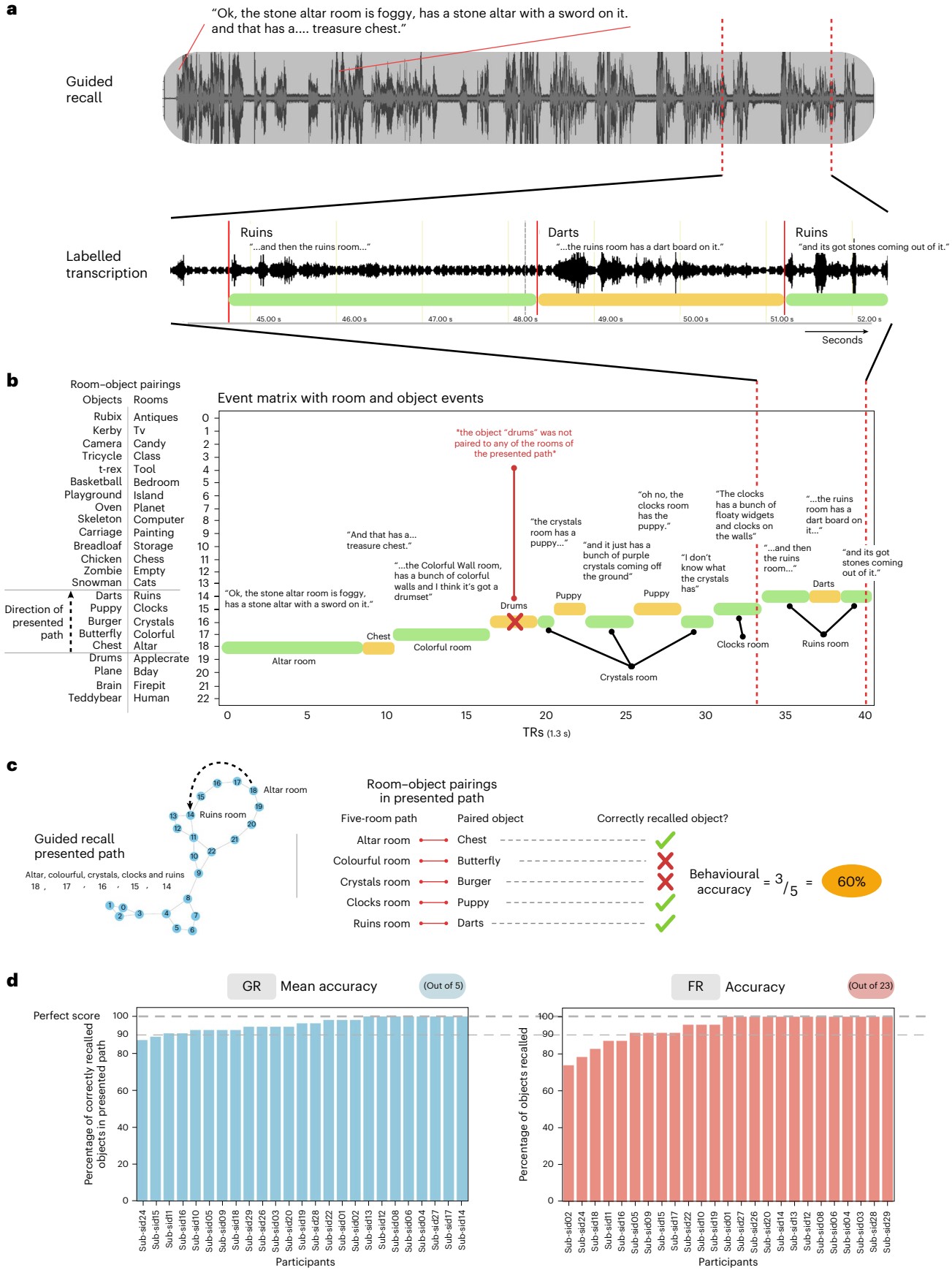

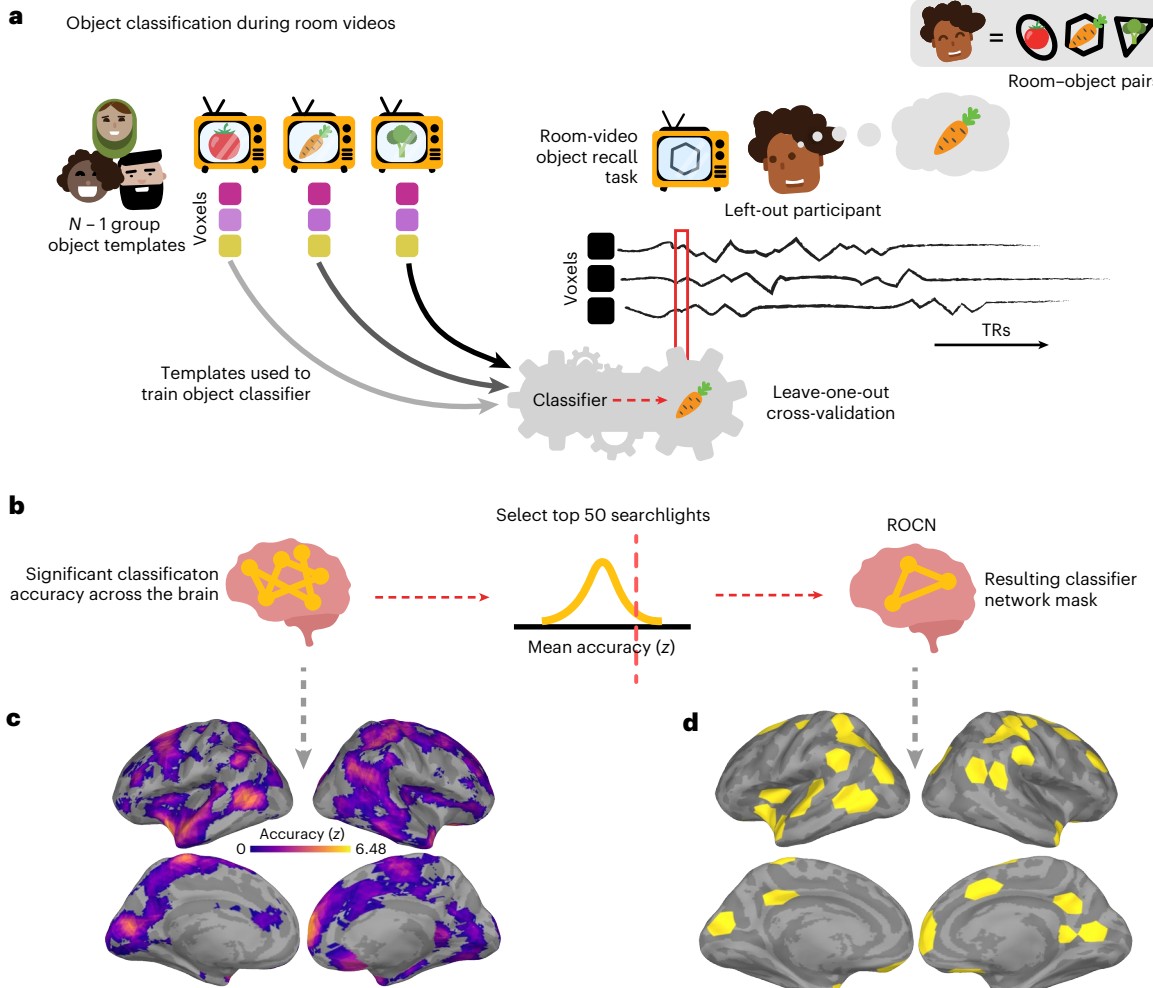

**Fig. 4 | ROCN methodology and surface maps. a**, During the postlearning room–video object recall task, participants watched a video of a room and verbally recalled the object that was paired to it. In a leave-one-participant-out cross-validation procedure, the characteristic object patterns of the $N-1$ group—evoked during a separate phase of the study in which participants viewed object videos—were used to train a multinomial logistic classifier. This classifier was then applied to each timepoint on the left-out participant's room–video object recall data. In the pictured example, the left-out participant, Fernando, is recalling the carrot object that was paired with the hexagon room currently being presented. The object classifier, trained on patterns evoked when other participants viewed the objects, was applied to each timepoint of Fernando's recall. We then measured the fraction of timepoints during the hexagon-room video that were classified as activating the carrot representation. **b**, For each searchlight, object classification accuracies for both room–video object recall videos for each participant were averaged together and then averaged across participants and z-scored relative to a null distribution. The 50 top-performing searchlights were then selected to form the ROCN. **c**, Average object classification accuracy during room–video object recall. The colour map shows the relative classification accuracy across all searchlights (thresholded to show only searchlights with above-chance accuracy). **d**, ROCN. The top 50 searchlights that were most sensitive to object reinstatement (yellow) were defined as the ROCN for subsequent analyses.

with a positive relationship between POCN reinstatement evidence and room reliability in the parietal cortex, superior frontal gyrus, insula, posterior medial cortex and dorsal occipital cortex. Across both recall tasks, there was a participant-specific benefit of room reliability in the posterior parietal cortex, posterior medial cortex, right insula and portions of the right lateral superior and middle frontal gyrus (Supplementary Fig. 6; refer to Supplementary Fig. 7 for guided and free recalls separately)

## Discussion

In this study, we posited that a cognitive map of spatial contexts is most useful as a container for future memories when locations have reliable representations, providing specific and consistent cues every time they are accessed. To test how the neural properties of a spatial context memory support new memories, we developed a paradigm that allowed us to quantify the within-participant reliability of a spatial context memory before it became the location in which a

new memory was formed, and then used this measure to predict the extent to which that new memory was remembered. We did this by having participants develop spatial context memories of a 23-room immersive VR memory palace, scanning them to extract the neural properties of their spatial memories for 'empty' rooms within the palace (prelearning phase) and then scanning them again afterwards, as they verbally recalled the 'filled' rooms and the objects that filled them (postlearning phase). We found that prelearning room reliability—the representational quality of an 'empty' memory scaffold—was predictive of postlearning object reinstatement in two types of verbal recall. We further showed that, in some regions, a participant's idiosyncratic room reliability values provided a predictive advantage beyond what could be inferred from room reliability patterns shared across participants. Finally, we showed that this relationship between room reliability and object reinstatement persists even after statistically controlling for room reinstatement at recall. By ruling out the alternative hypothesis that fluctuations in room reinstatement are

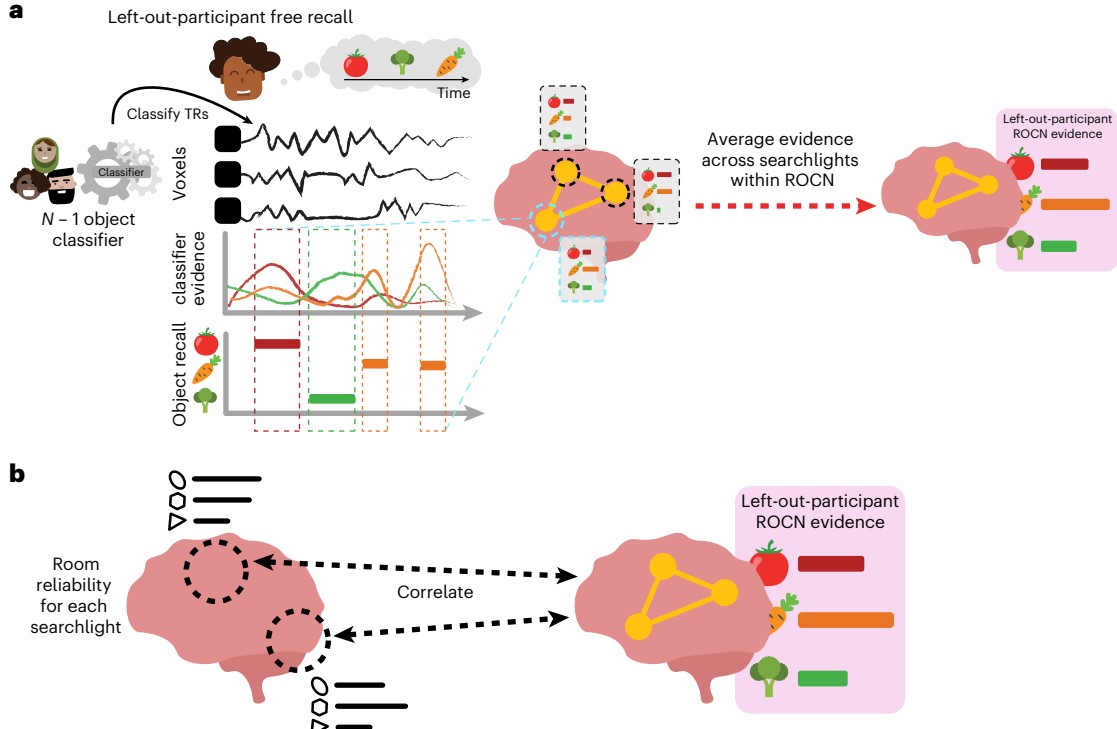

**Fig. 5 | Methodology for using room reliability to predict object reinstatement. a**, Illustration of methodology for how ROCN object reinstatement evidence was calculated from guided and free recalls. A leave-one-participant-out cross-validation procedure was used with a multinomial logistic classifier to predict object patterns at every timepoint of the left-out participant's recalls. To extract a single composite score of reinstatement evidence within the ROCN for every object and every participant, the classifier evidence for each object recalled was averaged within the ROCN mask across the timepoints when each object was verbally recalled. This yielded a single score for each object and each participant that represented the average object reinstatement evidence in the ROCN during guided or free recall. **b**, Illustration of methodology for how object reinstatement evidence was predicted by room reliability. In a searchlight analysis, reliability for a room (in that searchlight) was correlated with the corresponding composite score of reinstatement evidence (within the ROCN mask) for the object paired to that room. This correlation was computed across room–object pairs within each participant, and then those correlations were averaged across participants and, finally, across recall task types.

(fully) driving the effect, this control analysis provides indirect evidence in support of our preferred hypothesis—namely, that reliable room representations scaffold memory for objects by facilitating the binding of objects to rooms at encoding.

Theories in cognitive psychology have long argued that we develop knowledge structures that help to organize new information during encoding and later serve as a scaffold to recall specific details[29]; for example, prior work has discussed how event schemas[30], which describe the prototypical sequence of events associated with well-learned experiences (for example, restaurant visits), can support memory for new life events. In a similar fashion, knowledge about the structure and affordances of a spatial context can scaffold memories for experiences that occur in that context[6,31]. Our results support this general framework but also argue that all schematic containers are not equally effective at organizing memories; contexts that are only weakly learned and/or suffer interference from other contexts will not be effective scaffolds, consistent with work showing that repeated exposure to a single room versus distributed exposure to many rooms creates a more effective contextual cue[32]. In addition, our findings here also provide further support on the utility of VR as a tool for studying how spatial contexts can shape memory and behaviour[33].

## Room reliability is predictive of object reinstatement

There are two important features that make this study uniquely placed to investigate the role of spatial context scaffolds in episodic memory. First, the virtual rooms in this study are experienced in immersive VR and vary widely along many dimensions (room size and geometry, decoration, background soundtrack and so on), allowing participants to create rich and unique representations of individual rooms. Second, unlike other studies, neural patterns for each of the spatial contexts were acquired before the key learning event took place (here, the newly placed object in each location). These two features provided us with the opportunity to relate the neural patterns for 'empty' spatial contexts with the reinstatement of the objects that had been placed in them in a subsequent part of the experiment.

Specifically, our paradigm allowed us to relate the reliability of a room representation (the 'empty' scaffold) across the cortex to the reinstatement of the objects that had been placed in rooms explored in VR. In general, we found that object reinstatement was predicted by room reliability in the precuneus, insula, frontal cortex and regions throughout lateral parietal cortex (Fig. 6), suggesting that measuring the structural integrity of a spatial context representation before a life episode is predictive of how well that episode will be reinstated later. Moreover, these effects were found separately for both guided and free recall, providing an internal replication of our results and suggesting that stable context representations are useful for retrieval across multiple kinds of memory tasks. We observed strong effects in regions that are well known to support mental and virtual navigation[34–42], including the precuneus and the dorsal occipital lobe. Similar regions have also been identified in many types of tasks involving spatial knowledge: during spatially cued retrieval of real or imagined autobiographical memories[15,18,43], during recognition or retrieval of the spatial context in which an item was encountered[44–46], during the recollection of spatial relationships in two and three dimensions[47–50], during reinstatement of spatial contexts during item retrieval[51] and during the encoding and retrieval of items bound to a spatial context[52,53].

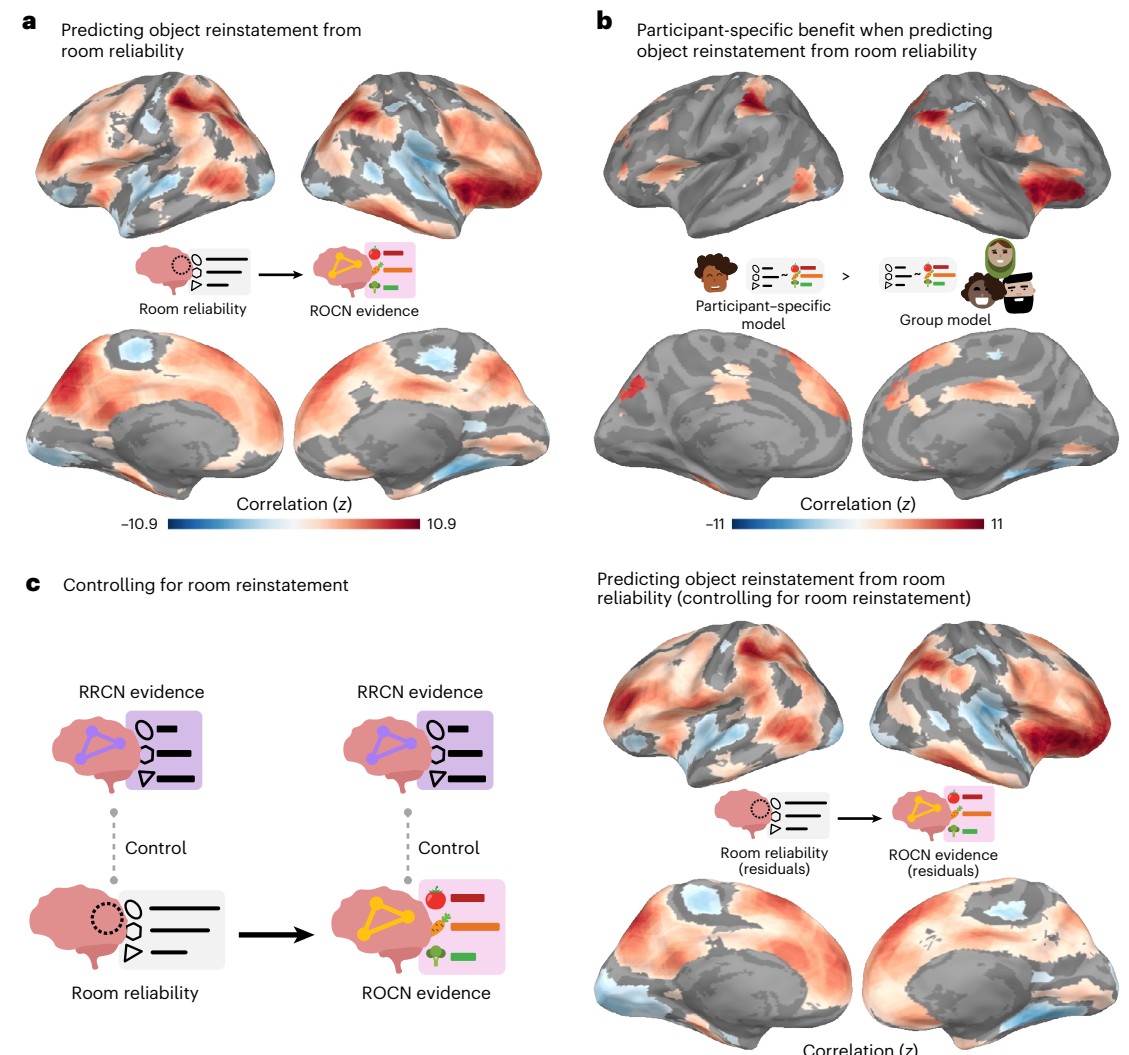

**a** Predicting object reinstatement from room reliability

Room reliability → ROCN evidence

Correlation (z)
−10.9 — 10.9

**b** Participant-specific benefit when predicting object reinstatement from room reliability

Participant–specific model > Group model

Correlation (z)
−11 — 11

**c** Controlling for room reinstatement

RRCN evidence          RRCN evidence

Control                Control

Room reliability → ROCN evidence

Predicting object reinstatement from room reliability (controlling for room reinstatement)

Room reliability (residuals) → ROCN evidence (residuals)

Correlation (z)
−11.4 — 11.4

**Fig. 6 | Predicting ROCN object reinstatement from room reliability.**
**a**, Relationship between ROCN object reinstatement and room reliability. Shown are regions where room reliability predicted ROCN object reinstatement across both guided and free recall. Objects placed in rooms with the most prelearning neural stability in these regions were reinstated the most strongly during retrieval. **b**, Model comparison results. Shown are regions in which room reliability predicted ROCN object reinstatement across both guided and free recall and where participant-specific room reliability provided additional predictive value. In these regions, the rooms that were most reliable for a specific participant (rather than rooms that were generally reliable across the group) were predictive of object recall for that specific participant. The surface maps presented in **b** show the intersection of the participant-specific models shown in **a** and the regions where there was a significant positive difference in the coefficient of determination between the original participant-specific model and the $N − 1$ group model. Statistical significance for the differences between

the coefficients of determination was determined by comparing the differences with a null distribution and FDR-correcting for $q < 0.05$. **c**, Controlling for room reinstatement. Left column: schematic illustrating how room reinstatement evidence in RRCN (during timepoints in which participants verbally recalled a room or its paired object) was regressed out of room reliability and ROCN object reinstatement scores. Room reliability residuals were then correlated at each searchlight with ROCN object reinstatement residuals. Right column: regions where room reliability predicted ROCN object reinstatement after controlling for room reinstatement during room and object recall. The surface maps presented in **a** and **c** were statistically thresholded by comparing correlations to a null distribution and then FDR-correcting for $q < 0.05$. All three surface maps are coloured based on the magnitude of the $z$-scored correlation values of the participant-specific model, with blue showing negative and red showing positive relationships.

Although these studies highlight the importance of spatial knowledge in a diverse range of learning and memory tasks, most of these studies focused on univariate or functional connectivity changes during the tasks, with few leveraging multivoxel pattern analyses (for example, refs. 15,51,54), and none quantifying the quality of the specific spatial representations used in these tasks. Thus, our work here, in combination with these prior studies, adds to the vast literature on spatial memory and provides a potential prerequisite for the successful completion of any spatial task: spatial context representations need to be reliable to be useful for subsequent memory storage.

In some other brain regions, we observed that room reliability in those regions was negatively related to subsequent object reinstatement. How can we explain these negative relationships? Because these regions are primarily in lower-level auditory and visual cortex, one possibility is that these regions code for lower-level sensory features, not spatial contexts, and the room reliability observed in these regions was actually a measure of how strongly these sensory properties were being represented. In this case, stronger representation of isolated features could be at odds with larger-scale and gist-like representations of the room geometry and semantic properties, making a room

less useful as a contextual anchor for subsequent object memory. Further work investigating object representation in the brain and its relationship to room reliability is required to aid in parsing the negative relationships we found.

What underlying mechanisms explain the relationship between object reinstatement and room reliability? Our hypothesis was that reliable room representations scaffold memory for objects by facilitating the binding of objects to rooms at encoding. Our finding that room reliability (measured before encoding) correlates with object reinstatement (measured during recall) is compatible with this 'facilitated binding at encoding' hypothesis. An alternative possibility is that successful reinstatement of reliable rooms during recall promotes object reinstatement for these rooms; this could give rise to a correlation between room reliability and object reinstatement, even in the absence of facilitated binding at encoding. We addressed this alternative hypothesis by controlling for room reinstatement during verbal recall and found that the relationship between room reliability and object reinstatement remained significant; furthermore, there were no areas that showed a significant decrease in the size of the effect when we controlled for room reinstatement. The results of this control analysis provide indirect support for our hypothesis that room reliability supports improved room–object binding at encoding; namely, a reliable spatial context representation may provide a stable schematic map that facilitates the integration of new episodic content—the more reliable the container, the easier it is to populate it with information. Future work in which participants are scanned during object–location encoding would help shed additional light on how room reliability enhances the creation of episodic memories.

### Room reliability

We described the representational stability and distinctiveness of a spatial context through a reliability score that measured the specificity of a room's representation across runs. These spatial contexts were designed to be visually and auditorily rich to reflect real-world contexts. Given that room reliability was derived from audiovisual stimuli, it was not surprising to find the strongest reliability in the visual and auditory cortex. In addition to these sensory regions, we found significant room reliability in other regions that have been implicated in higher-level processing: the parietal cortex (including the intraparietal sulcus), posterior medial cortex (including the precuneus) and lateral prefrontal cortex (including the premotor cortex). In other studies, these regions have been shown to maintain specific scenes or events within stories along various timescales during movie watching[55–59]. These regions may help to ensure stable and distinctive representations of the high-level properties of the current situation that go beyond low-level sensory properties—an idea consistent with prior work showing that these regions represent event types shared across stories, regardless of whether the story is presented as an audio narrative or an audiovisual movie[59,60]. Although some of this event structure can arise from the temporal dynamics of the stimulus itself, internal schemas can also be used to actively organize an experience into stable events[61]. Our results suggest that this kind of top-down stabilization may be most effective when the schema itself is highly reliable, providing a robust starting point for building episodic event representations.

Although high pattern similarity across identical trials is related to better subsequent memory[62], purposefully increasing variability in item encoding by varying the encoding context has been shown to improve item memory[63,64], perhaps by increasing the number of possible retrieval cues for the item (see, for example, refs. 65,66). It is therefore possible that there are some situations in which unstable context representations would be useful for creating memories, for example, if items are studied multiple times in a context and then recognition memory is tested in a novel context. However, in our paradigm, participants were explicitly using a context-based strategy for retrieving items, mentally simulating rooms and trajectories through rooms to reinstate item memories. In this case, we would expect that having a reliable contextual index for episodic memories would be critical for effective recall of items, consistent with our findings that stability in scene-related brain regions predicted item reinstatement. Future work could investigate whether this relationship disappears or reverses in other situations, such as when many items are paired with the same room (reducing the usefulness of rooms as memory cues), or when rooms have features that vary, for example, with time of day (such that representational variability might reflect meaningful changes in contextual features), or when the recall task requires reporting only objects while suppressing recall of room features. Similarly, novelty may influence how room reliability scaffolds memory: a new context may be less stable than a highly familiar location but could still enhance memory because its novelty promotes additional attention and processing. Future work examining how repeated exposure and contextual novelty interact with reliability could shed new light on their contributions to memory.

### Our experimental paradigm and the method of loci

Our 'memory palace' paradigm draws inspiration from the mnemonic technique called the method of loci (MOL), in which items are associated with an imagined sequence of spatial locations in a prelearned map. However, our study diverges from this technique in several key ways. Unlike many implementations of MOL, participants were not required to encode or recall to-be-remembered items in an explicit linear sequence of rooms, nor were they instructed to use any particular mnemonic during room–object binding. Instead, participants explored the virtual environment freely and developed their own strategies for memorization.

Despite these differences, the motivation for this technique is related to the hypothesis tested in this study: that a well-learned spatial map consisting of many distinct locations is the optimal encoding environment for new item memories. The learnability of this technique suggests that it may rely on inherent spatial memory structures shared across people. In fact, the ability to improve memory through this spatially based technique has been shown across multiple studies behaviourally and neurally (behavioural[26,67–70], among many; neural[52,54,71–77]). Generally, neuroimaging studies of this technique have largely focused on the impact of MOL (at varying levels of training or compared with other mnemonics) during item encoding[71–74], with only a few performing univariate contrasts during recall[52,75–77], and only one, to our knowledge, examining multivariate pattern activity for loci, items and their conjunctive associations[54]. The univariate results during recall have shown enhanced engagement of regions including retrosplenial cortex and precuneus after instruction in MOL[52], suggesting that spatial representations of loci are strategically activated during retrieval. A recent study measuring multivariate activity patterns during MOL[54] found robust representations for individual loci during the creation and retrieval of item–locus pairs in regions including the precuneus and posterior parietal cortex, suggesting potentially overlapping mechanisms in how our naive participants and MOL-trained individuals use spatial information for item memorization. It remains an open question whether enhanced room reliability helps support memorization when using MOL.

## Conclusion

After participants learned a complex spatial environment in VR, we measured the neural reliability of each spatial location within this map. When participants later used this environment to learn a new set of room–object associations, we showed that this room reliability measure could predict the degree to which objects associated with each room successfully came online during naturalistic recall. Together, these results showcase how the quality of a spatial context can be quantified and used to 'audit' its utility as a memory scaffold for future memory.

## Methods

### Participants

Data were collected from a total of 30 participants between the ages of 21 and 32 years (16 females, 14 males) with normal or corrected-to-normal visual acuity. At the end of the study, participants were paid and debriefed about the purpose of the study. Every effort was made to recruit an equal number of female and male participants and to ensure that minorities were represented in proportion to the composition of the local community. The experimental protocol was approved by the institutional review board (IRB) of Princeton University, and all participants provided their written informed consent (IRB #7225). Due to technical difficulties (corrupted and missing files), 5 participants were excluded, leaving a total of 25 participants (11 females, 14 males).

### Stimuli

**VR stimuli.** *Environment.* A custom-built VR environment made up of 23 interconnected distinct rooms with distinct soundtracks was explored by participants using a head-mounted VR display. Each of the rooms was built to be as visually and aurally distinct as possible. To that end, for visual distinctiveness, each room followed a different theme (for example, planetarium and computer store) with theme-congruent objects carefully placed throughout, and the rooms had different shapes (for example, oval and rectangle) and different sizes (for example, large and small). To promote auditory distinctiveness, each room had a distinct soundtrack on loop that was audible only when a participant entered each room and some rooms contained specific sound effects that matched the room context (for example, bird chirps if the room had a window facing the outside).

The majority of rooms were connected with only two other connecting rooms, while a few, 'hubs', had more than two connecting rooms. Among all 23 rooms, 16 rooms (70%) were connected with 2 other rooms, 6 rooms (26%) were connected with 3 other rooms and 1 room (4%) was connected with 4 other rooms.

To reduce the potential for motion sickness, participants explored the virtual world while seated in a 360°-rotatable chair, and any instance of participant-initiated teleportation was followed by a short and smooth fade-in-and-out of black. Participants teleported within and between rooms by pressing a button on a wireless controller that would appear digitally reconstructed in VR as a three-dimensional (3D) object. The range of teleportation was limited to force teleportation across small distances and to avoid fast teleportation across rooms. Rooms were connected by doorways; given the current room a participant was in, only the immediately connected rooms were visually accessible via the doorways, while further-away rooms were culled from view.

*Music and sounds.* Sounds of birds, ambience, firewood crackling and others were manually recorded or freely downloaded from the internet. Music for each room was either custom-composed in Ableton Live software, downloaded from the internet or requested from professional composers (Supplementary Table 1).

All tasks were presented on a wired HTC Vive head-mounted display (1,080 × 1,200 resolution per eye, with a 90-Hz refresh rate, built-in headphones and integrated microphone), which was connected with a wire to a computer running 64-bit Windows 10 on an Intel Core i7-6800K CPU @ 3.40 GHz with 32 GB random access memory and an Nvidia GeForce GTX 1080 graphics card.

All tasks and visual presentations were created and coded in Unity3D 5.5.2f1 (and 2017.1.2f1), a game-development platform, with Virtual Reality Toolkit (VRTK; vrtk.io), a virtual-reality programming tool kit for Unity3D. The majority of 3D models, textures, environments and other assets were custom-built using SketchUp (sketchup.com) or Blender (blender.org). The remaining assets were downloaded from the Unity Asset Store (assetstore.unity.com), Turbosquid (turbosquid.com) or other publicly available online repositories and then modified using Blender to reduce model complexity and size.

**Scanning stimuli.** During scanning, participants were presented with videos of rooms and videos of objects. These videos were generated beforehand and presented to participants in a pseudorandom order.

*Room videos.* To generate the room videos using Unity, a virtual camera was placed in the centre of each room. The camera was scripted to rotate a full 360° to capture the panorama of each room within 10 s. OBS Studio (obsproject.com) was used to screen capture the output of the virtual camera. Each room video lasted 10 s and was followed by a 5-s interstimulus interval before the next video.

*Object videos.* To generate the object videos, a virtual photography studio was created with a blank backdrop and a 3-point lighting set-up. All 23 objects were placed in the centre of the virtual studio and scripted to rotate 360° in front of a virtual camera facing them within 10 s. OBS studio was used to screen capture the output of the virtual camera. Similarly to the room videos, each object video lasted 10 s and was followed by a 5-s interstimulus interval before the next video.

*Stimulus presentation.* All generated stimuli were presented to participants in the scanner using PsychoPy[78] to time task and stimulus presentations with the scanner trigger. Every presented video or task instruction was preceded by a 5-s black screen.

### Data acquisition and preprocessing

**MRI acquisition and preprocessing.** MRI data were collected on a 3T full-body scanner (Siemens Prisma) with a 64-channel head coil. Functional images were acquired using an interleaved multiband echo-planar imaging (EPI) sequence (repetition time (TR) 1,300 ms, echo time (TE) 33 ms, flip angle 80°, whole-brain coverage, 2 mm slice thickness, field of view (FOV) 192 mm$^2$, simultaneous multislice (SMS) factor 4). Anatomical images were acquired using a T1-weighted (T1w) magnetization-prepared rapid-acquisition gradient echo (MPRAGE) pulse sequence (1 mm$^3$ resolution). Anatomical images were acquired in a 6-min scan before the functional scans; during this scan, participants watched videos of paragliding from YouTube. Field maps were collected but not used in our preprocessing pipeline.

All raw data acquired from MRI were converted to BIDS formatting (BIDS version 1.0.1), anatomical images were defaced using pydeface (version 2.0.0) and resulting data were subsequently preprocessed using fMRIPrep version 1.0.3, a Nipype[79,80]-based tool. Each T1w volume was corrected for intensity non-uniformity using N4BiasFieldCorrection v2.1.0[81] and skull-stripped using antsBrainExtraction.sh v2.1.0 (using the OASIS template). Brain surfaces were reconstructed using recon-all from FreeSurfer v6.0.0[82], and the brain mask estimated previously was refined with a custom variation of the method to reconcile cortical gray matter segmentations derived from Advanced Normalization Tools (ANTs) and FreeSurfer, as implemented in Mindboggle[83]. Volume-based spatial normalization to the ICBM152 Nonlinear Asymmetrical template version 2009c[84] was performed through nonlinear registration with the antsRegistration tool of ANTs v2.1.0[85], using brain-extracted versions of both T1w volume and template. Brain tissue segmentation of cerebrospinal fluid (CSF), white matter and grey matter was performed on the brain-extracted T1w using fast[86] (FSL v5.0.9). Surface-based normalization based on nonlinear registration of sulcal curvature was applied using the fsaverage6 surface template from FreeSurfer.

Functional data were slice time corrected using 3dTshift from AFNI v16.2.07[87] and motion corrected using mcflirt (FSL v5.0.9[88]). 'Fieldmap-less' distortion correction was performed by coregistering the functional image to the same-participant T1w image with intensity inverted[89,90], constrained with an average fieldmap template[91], implemented with antsRegistration (ANTs). This was followed by coregistration to the corresponding T1w using boundary-based registration[92] with nine degrees of freedom, using bbregister (FreeSurfer v6.0.0). Motion-correcting transformations, field distortion

correcting warp, blood oxygenation level-dependent (BOLD)-to-T1w transformation and T1w-to-template (MNI) warp were concatenated and applied in a single step using antsApplyTransforms (ANTs v2.1.0) using Lanczos interpolation.

Physiological noise regressors were extracted applying CompCor[93]. Principal components were estimated for the two CompCor variants: temporal (tCompCor) and anatomical (aCompCor). A mask to exclude signal with cortical origin was obtained by eroding the brain mask, ensuring it contained only subcortical structures. Six tCompCor components were then calculated including only the top 5% variable voxels within that subcortical mask. For aCompCor, six components were calculated within the intersection of the subcortical mask and the union of CSF and white matter masks calculated in T1w space, after their projection to the native space of each functional run. Framewise displacement[94] was calculated for each functional run using the implementation of Nipype.

**Additional preprocessing.** After fMRI data were aligned and pre-processed to fsaverage6 resampling, the resampled data were further preprocessed using a custom Python script that removed nuisance regressors, including the six degrees of freedom motion correction estimates; framewise displacement (the estimated bulk head motion); head motion estimates from white matter and CSF; and cosine bases for high-pass filtering to account for low-frequency signal drifts (up to 0.008 Hz, or 125 s). Within the same Python script, the resulting timeseries data were z-scored for each run (that is, task), such that there was a single preprocessed timeseries per task (for example, prelearning room videos, postlearning object videos, recall and so on).

## Experimental paradigm

The study took place on two consecutive days and was composed of a behavioural session on day 1 and a behavioural and two scanning sessions on day 2.

### Day 1

On day 1, participants were familiarized with the virtual environment and exposed to two VR foraging games and hand-drawing tasks to facilitate the learning of the spatial layout. Specifically, on day 1, after participants read and signed the consent and screening documents, participants were informed about what they would be experiencing in VR and about the safety measures taken to ensure their safety and comfort. They were told that they would be seated to decrease potential dizziness that arises more commonly during VR that involves standing. They were also informed that at any time the experiment could be stopped if they are feeling uncomfortable or dizzy. They were told that they would play two foraging games in VR that involve freely moving through the VR environment with the goal of collecting floating cubes. In the first game, they had to collect a cube from every room. In the second game, they had to repeatedly navigate to designated rooms to collect additional cubes. They played the second game twice. Between each game, participants were asked to draw a bird's-eye-view map based on their current knowledge of the environment (Supplementary Fig. 1). We did this to ensure participants were learning the spatial layout of the environment. By the end of the behavioural session, participants had completed a total of three games and three maps. Throughout the experimental session, the experimenter checked on the participant's overall comfort and reminded them that if they felt dizzy or nauseous, the experiment could be stopped at any time without consequence. After the completion of the foraging tasks, the participants were compensated and reminded to return the next day for the two scanning sessions and the additional VR behavioural session.

### Day 2

On day 2 (1 day later), three sessions took place: In the first session, participants were scanned with fMRI for a small battery of encoding tasks (prelearning scan); in the second session, participants learned room–object associations in VR for randomly placed objects in each of the 23 rooms (learning behavioural session), and in the third session,

participants were scanned again with fMRI as they proceeded through a battery of encoding and retrieval tasks (postlearning scan).

Session 1 (prelearning scan): On day 2, participants were greeted at the MRI room, asked to draw a bird's-eye-view map of the environment (as had been done the day before). After listening to a short unrelated audio clip in the scanner to verify volume level, participants were told that they would be presented with two sets of audiovisual stimuli of the rooms. In the first set they saw 360° room rotation videos of all the rooms (that is, prelearning room videos) and were instructed to verbally recall the name of the room when they recognized it. The second set, which was viewed after the first, was exactly the same as the first, except the room order was randomized for each participant. Every stimulus presentation was preceded by a 5-s blank screen.

Session 2 (learning behavioural session): After participants finished the prelearning scan, they were taken out of the scanner bore and instructed to carefully stand up. They were then guided back to the behavioural room with the VR equipment to complete the second session of VR. In this session, participants were refamiliarized with the environment by playing the first foraging game again. Afterwards, they drew a bird's-eye-view map once again and then were told that, when they returned to the virtual world, they would find 23 different 3D objects scattered in each of the 23 rooms. They were then given 15 min to memorize the room–object pairings.

Session 3 (postlearning scan): After the 15 min that participants were given to memorize the room–object pairings had elapsed, participants were guided back into the MRI room. Before getting into the scanner, participants were told that they would be asked to verbally recall in as much detail as possible the 23 room–object pairings. They were also told that they would be presented with the same audiovisual stimuli from session 1, and they would also view an additional set of videos that included objects. In the first task (free recall), participants were asked to describe in as much detail as possible all the rooms and objects that they saw in VR. In the second task (guided recall), participants were asked to recall with as much detail as possible the appearance of the rooms and objects along specific five-room paths within the environment. The names of the five rooms were visible on screen. They did this guided recall task 11 times, each time with a different five-room path. When they had completed recalling the rooms and objects to the best of their ability for the free recall and guided recall tasks, they were told to inform the experimenter by saying 'done'. In the third task (which we label as room–video object recall), participants were exposed to the same 360° room rotation videos from the aforementioned prelearning room video tasks, but this time, when they were shown a room video, they were tasked to recall the novel object that had been placed in it (that is, room–video object recall). They did this task twice for all rooms. Because room–object pairings were generated randomly for each participant, the objects recalled during this task were usually different across participants. Afterwards, in the fourth in-scanner task, participants saw the postlearning object videos. During these, participants performed the object–video room recall tasks: participants were shown 360° object rotation videos and instructed to say the name of the room that was paired with that object. They did this task twice for all objects.

## Searchlights

Our searchlights were generated by constructing them with every valid vertex as their centre, then iteratively removing the most-redundant searchlights until no more could be removed while covering each vertex with at least ten searchlights. This process yielded 1,483 searchlights per hemisphere.

## Hippocampus

Our full hippocampus ROI was extracted from a freesurfer subcortical parcellation. This ROI was then split into an anterior portion ($y > -20$) and posterior portion ($y \leq -20$) in MNI space[59,95,96].

## Behaviour

**Behavioural event matrices.** *Prelearning and postlearning room, and object videos.* The timing of stimulus presentations for every room and object was logged, and a custom Python script was used to convert the timestamps to a behavioural timeseries event matrix that marked the start and end of every stimulus presentation for every participant. The resulting matrix that contained the timing (in milliseconds) and room or object identity was then downsampled to 1.3-s TRs and used in subsequent analyses to index into a participant's BOLD timeseries data to identify the moments in time participants were encoding a specific video. In sum, the Python script generated six different behavioural event matrices, two prelearning room event matrices (that is, prelearning room videos), two postlearning room event matrices (that is, room–video object recall tasks) and two postlearning object event matrices (that is, object–video room recall tasks).

*Postlearning free recall and guided recall.* Participants were asked to recall and describe the rooms in the virtual environment and the objects paired to the rooms. Using TotalRecall (https://memory.psych.upenn.edu/TotalRecall), audio files of participant's recalls were imported and transcribed by timestamping the start of a room or object verbal description. For example, if a participant said, "I remember walking through the chess room, it had large chess pieces. The object in there was a basketball…", the start and end of the 'chess room' timestamps would have been at the start and end of the first sentence, respectively. This is because we assumed that the room would have come to mind at the start of the sentence rather than midway. Similarly, the object start timestamp would have been considered the start of the second sentence. For every participant, these timestamps were then imported into a custom Python script that generated a behavioural timeseries event matrix that marked the start and end of each verbal room or object recall. This resulted in 11 guided recall and 1 free recall behavioural timeseries event matrices that indicated the trajectory of room or object recalls. These were then downsampled to 1.3-s TRs and used in subsequent analyses to index into a participant's BOLD timeseries data to identify the moments in time a participant was recalling a particular room or object.

*Time spent recalling rooms or objects.* To assess whether certain rooms or objects were discussed significantly more than others during recall, we conducted an across-participant global mean comparison. For each room and object, we computed the mean time spent speaking across participants. We then performed a one-sample *t*-test for each room (or object), testing whether its average recall time significantly deviated from the grand mean (that is, the average across all rooms or objects). Next, we applied a Bonferroni correction to the resulting *P* values to account for multiple comparisons.

*Contiguity in free recall.* To assess whether participants tended to recall spatially connected rooms in sequence, we computed, for each participant, the proportion of times each room transition was to an adjacent room (that is, graph distance of 1 in the adjacency matrix of the virtual environment). Self-loops, where the same room was recalled consecutively, were excluded. To calculate the baseline probability that a participant may have recalled an adjacent room just by chance, for each transition, we counted the number of currently adjacent rooms divided by the 22 possible other rooms (excluding the current room) and then averaged across all transitions. To test for significance, we ran a paired-sample *t*-test where we compared each participant's proportion of contiguous recall with their chance baseline (Supplementary Fig. 3f).

## fMRI analysis

**Characteristic object patterns.** To acquire the characteristic neural patterns for objects ('object templates') we created 23 regressors to model the neural response to each of the 23 objects. We placed each of the 23 object regressors in a design matrix that marked the transitions between object videos across both postlearning object video tasks; the matrix was convolved with a haemodynamic response function (HRF) from AFNI (Cox, 1996) and then *z*-scored. We then extracted the characteristic spatial pattern across vertices for each object by fitting a general linear model (within each participant) to the timeseries of each vertex using these 23 regressors. Doing this simultaneously across both postlearning object videos yielded a single set of 23 characteristic object spatial patterns across vertices for each participant. These object templates, which were obtained for every participant, were then used in subsequent analyses for training multinomial logistic classifiers. All object classifiers described in this Article were trained on these perception-evoked patterns.

**Characteristic room patterns.** To acquire the characteristic neural patterns for rooms ('room templates'), we followed the same procedure that we used for extracting object templates, but here—instead of using postlearning object videos—we used the prelearning room videos obtained from the first scanning session on day 2 to obtain the characteristic spatial pattern across vertices for every room.

**Room reliability.** We hypothesized that, for a room to serve as an effective retrieval cue for associated memories (that is, objects paired to rooms), the neural representation for that room must be stable over time and distinct from other room patterns. We captured these properties with a composite measure we called room reliability. Crucially, this measure was computed based on data that were collected before participants learning the room–object associations. This ensured that our room templates, and therefore our room reliability measure, were not confounded with object information.

To compute room reliability, we obtained the characteristic spatial pattern for each room for each participant, using the procedure outlined above (in the 'Characteristic object patterns' section), but for room videos instead of object videos. Doing this for both prelearning room video tasks yielded 2 sets of 23 characteristic spatial patterns across vertices (separated in time) for each participant.

We then created a room pattern similarity matrix by correlating the characteristic neural patterns for the rooms from the first prelearning room video set with the neural patterns for the rooms from the second set. This yielded a 23 × 23 correlation (similarity) matrix for each participant. Because the two prelearning room videos were separated by a delay, the principal diagonal indicated the similarity of the room representations over time—this was our measure of the stability of the room representations. Similarly, the off-diagonal entries indicated the similarity of one room to another over time, reflecting greater distinctiveness. To create our composite room reliability score for each room, we subtracted the average similarity of the off-diagonal entries (how similar room A is to other rooms over time) from the principal diagonal entry (how similar room A is to itself over time). A large positive difference indicated that a particular room (for example, room A) was more similar to itself over time than it was to other rooms, indicating its stability and its distinctiveness from other rooms. We did this procedure to obtain a room reliability score for each room of each participant. To quantify significance, for each participant, we averaged reliability across all rooms to get a single difference score per vertex, and performed a one-sample *t*-test on these differences against zero before running false discovery rate (FDR) correction on the resulting *P* values and thresholding at $q < 0.05$.

**Object classifier network selection.** To identify which regions across the brain are involved in the retrieval of object information during guided or free recall, we first needed to identify regions across the brain that could discriminate between objects. To do this, we used two separate phases of the experiment to extract networks that could classify objects during retrieval (when perceptual details of an object

were not available) and during perception (when the perceptual details of an object were available). After participants had learned the room–object associations in VR, they were scanned while they watched videos of rooms and asked to recall the objects that were in them (room–video object recall task/postlearning room videos). We used this cued-recall task to identify the retrieval networks (ROCN) involved in classifying objects during room videos. Similarly, we identified the networks (POCN) involved in perception of objects, by classifying objects during postlearning object videos. Importantly, to avoid circularity in our analyses, all object classifiers (whether those made for ROCN or POCN) were trained with $N-1$ object perception data using a leave-one-participant-out procedure 25 times, where testing occurred on the left-out participant. The fact that each of the 24 participants in the training dataset had their own set of random room–object pairings ensured that the classifier was able to learn object representations that were not contaminated with room information (by contrast, if we had used a within-participant classification approach, room and object information would have been confounded, because objects were scanned only after they had been paired with a particular room). In other words, because room–object pairings were randomized for every participant, and object evidence for each participant was classified based on object templates derived from the other $N-1$ participants, any room-related information in the object templates would be unrelated to the room-related information in this left-out participant.

**Network selection procedure**

In brief, we ran object classifiers on postlearning room videos, where participants had been asked to recall the name of the object paired to the shown room, to identify a network of regions involved in retrieving non-visible object identity. This process involved the following steps: (1) acquiring the characteristic neural pattern for each object (postlearning object templates); (2) using a leave-one-participant-out multinomial logistic classifier, trained on the object template patterns for the $(N-1)$ group, to predict object identity in the excluded participant's postlearning room videos (to identify the ROCN) or postlearning object videos (to identify the POCN); and (3) averaging classifier performance (that is, accuracy) across all validation searchlights and then selecting the top 50 best classifier searchlights (~3%). This procedure was done on each searchlight plus the hippocampus ROIs for all participants. Further details are outlined below.

(1) Characteristic object patterns (object templates): To extract characteristic neural patterns for objects ('object templates') we used the procedure previously described in the 'Characteristic object patterns' section.

(2) Classifier cross-validation procedure: We applied a leave-one-out cross-validation procedure to predict the left-out participant's object reinstatement at every timepoint during postlearning room viewing after fitting (that is, training) a multinomial logistic classifier with the other participants' object pattern templates (that is, the characteristic spatial patterns estimated from the general linear model). More specifically, we shifted the left-out participant's postlearning room video's BOLD timeseries by four TRs to approximate the HRF delay and then trained the classifier with the other participants' object templates before predicting the object class for every timepoint of every room video. To assess the significance of classifier accuracy, we compared the classifier predictions with the correct object class labels and generated a null distribution of accuracies by shuffling the correct labels 1,000 times without replacement while preserving their temporal contiguity; this null distribution was used later to identify searchlights that had above-chance accuracy. We did this procedure across all participants such that every participant served as a test participant.

(3a) ROCN selection: Postlearning room videos were shown twice to each participant. We ran the leave-one-out cross-validation procedure described in the previous section for both runs of the postlearning room viewing separately and then, across all participants and both runs, averaged the classifier accuracy including the corresponding null

distributions. We then $z$-scored the searchlights' (and hippocampus ROIs') performance by comparing the true average accuracies to the average null distribution of accuracies. Afterwards, we extracted the top 50 ROIs with the highest $z$ scores. This resulted in 50 searchlights (distributed unevenly across hemispheres and excluding hippocampus) corresponding to the searchlights with the top performing classifier performance; these 50 searchlights made up the object retrieval network that we used as an ROI mask in subsequent analyses.

(3b) POCN: We applied the same procedure described in the 'ROCN selection' section, but instead of classifying non-visible object identity from postlearning room videos, we classified object identity from the postlearning object videos where objects were perceptually visible. In a similar fashion, we extracted the top 50 ROIs by sorting the $z$ score of accuracies to obtain the network involved in classifying visible objects. Unsurprisingly, this network was focused on primary visual cortex.

(3c) RRCN selection: We applied a similar procedure described in the 'ROCN selection' section, but instead of classifying non-visible object identity from postlearning room videos, we classified room identity from the postlearning object videos where objects (but not rooms) were perceptually visible. In a similar fashion, we extracted the top 50 ROIs by sorting the $z$ score of accuracies to obtain the network involved in classifying room memories. Importantly, the room classifiers were trained on the prelearning room template patterns for the $(N-1)$ group to predict the recalled room during the held-out participant's postlearning object videos. This ensured that (1) the held-out participant's own room templates were never used for testing, avoiding circularity and (2) the room templates of the group were sourced before any room–object associations were learned, eliminating the potential for these room templates to be contaminated by object information.

**Object evidence during guided and free recall.** We used the same leave-one-out cross-validation procedure described previously to predict object identity during guided and free recalls. As described previously, we shifted each recall timeseries (11 guided recalls and 1 free recall) by 4 TRs to approximate the HRF delay, and used the multinomial classifier to predict object classes at every timepoint for every participant's recalls. Given that that multiclass classifier was trained on all 23 object classes, we obtained a probability distribution across all 23 classes that described the evidence of each class being reinstated at each timepoint. For any specific guided or free recall, we collected the total object evidence across all timepoints when a participant verbally recalled that object, regardless of whether the associated room was also recalled. We did not condition our object reinstatement measure on recall of the correctly associated room because we were interested in studying how prelearning room reliability affects object recall in general (as opposed to studying how reinstatement of a room representation at recall triggers retrieval of the associated object). We then averaged these timepoints across recall runs (guided and free recalls separately). For example, if during the first guided recall a participant verbally recalled the object 'teddy bear' in two separate chunks of time for a total of 16 TRs, we collected the classifier probability for 'teddy bear' across those 16 TRs. We then did the same for every TR in which 'teddy bear' was recalled in all other guided recalls and averaged the results to obtain the total 'teddy bear' evidence. For a given participant, we did this for each object combining across all 11 guided recalls and, separately, for the participant's single free recall, yielding 23 mean object probabilities for each type of recall task (guided and free recall) for each searchlight.

We wanted to obtain a single value for each object in each participant (separately for guided and free recall), indicating how well that object was reinstated during recall. We did this in two ways: by averaging an object probability across all searchlights that were part of ROCN or POCN to obtain an overall ROCN or POCN reinstatement score, respectively, for each object and each participant. These scores were then used as our overall network object reinstatement measures in subsequent analyses.

**Relationship between room reliability and object reinstatement.**
We hypothesized that rooms with more reliable representations in the prelearning scans would be associated with higher levels of object reinstatement during self-paced verbal recall. To do this, we ran a searchlight analysis where we correlated the reliability of a room (see 'Room reliability' section) with the network's evidence for the object paired to that room (see 'Object classifier network selection' section). We did this for every room–object pair within a participant. For example, for a particular participant, the 23 room reliabilities were correlated with the corresponding 23 object reinstatement probabilities from the retrieval network. Afterwards, we averaged the Fisher-$z$-transformed correlations across participants and recall task types (that is, guided versus free recall) to generate a single composite correlation map. To test for statistical significance, we ran a non-parametric permutation test in which we randomly shuffled the object labels 1,000 times to generate a null distribution of correlations within participants and for both recall types. Significance testing was then performed using the combined null distribution, and resulting $P$ values were FDR-corrected (Fig. 6a). For reference, the results for each recall task type individually are presented in (Supplementary Fig. 7).

**Benefit of participant-specific room reliability.** The analysis shown in Fig. 6a assesses whether there is a within-participant relationship between room reliability (in a particular searchlight) and object reinstatement. Importantly, there are two possible explanations for this effect (not mutually exclusive). The first is that, within a particular participant, there are idiosyncratic differences in room reliability that predict object reinstatement for that participant; we call this a participant-specific effect. However, there is a second possible explanation: some rooms may be more reliable than others (averaging across the whole group), and these generally more reliable rooms may support better object reinstatement on average (for example, the chess room might consistently be better represented across people and support better object recall); we call this a group-wise effect. Both kinds of effect are important, but they have different connotations: if the relationship between room reliability and object reinstatement is driven by idiosyncratic (participant-specific) factors, then there is predictive value in doing a 'personalized audit' of the person's memory palace by scanning them; but if there is only a group-wise relationship, there is no need to collect scanning data from a new person, so long as you already have data on room reliability from the rest of the group. To assess whether the observed within-participants relationship between room reliability and object reinstatement has a participant-specific component, we compared the predictive performance of an ordinary least-squares regression derived from a participant's own room reliability values with one based on other individuals' reliability values. Specifically, for the participant-specific model (as in Fig. 6a), we calculated the coefficient of determination ($R^2$) of a model where the participant's object reinstatement probabilities were predicted by their own room reliability values. For the group-wise model, we iteratively predicted that participant's reinstatement probabilities from every single other participant and averaged the resulting $N-1$ $R^2$ values. We then ran a model comparison test where we took the difference between the $R^2$ of the participant-specific model and the average $R^2$ from the other-participant prediction models. A significant positive difference in this analysis indicates that the participant-specific model explains the variability in object evidence better than other individuals (and thus the observed results cannot be entirely due to the group-wise effect). To test for statistical significance, we ran a non-parametric permutation test where the object labels were randomly shuffled 1,000 times to generate a null distribution of model performance for each model. To generate a single composite map summarizing model performance across recall tasks, we averaged $R^2$ values across guided and free recalls for each model separately, computed the difference in $R^2$ and tested for significance using a non-parametric permutation test on the combined null distribution of $R^2$ differences, followed by FDR correction on the resulting $P$ values. For completeness, separate results for guided and free recall are provided in Supplementary Fig. 7, while the main results reflect the combined composite analysis (Fig. 6).

**Partial correlation analysis controlling for room reinstatement.** To test whether room reliability predicted subsequent object reinstatement when controlling for room reinstatement at recall, we conducted a partial correlation analysis. Specifically, we asked whether the correlation between room reliability and object reinstatement (Fig. 6a) remained significant after regressing room reinstatement at recall out of both of these other variables.

To do this, we first constructed a RRCN. As described in the 'Network selection procedure' section, we followed a similar approach to identify ROCN and POCN. In brief, we used a leave-one-participant-out cross-validation procedure in which we classified room recall during the held-out participant's perception of object videos. The top 50 best-performing searchlights were used to define the RRCN, which was then used as a mask to extract room reinstatement evidence for our partial correlation analysis.

In this analysis, we wanted to control for room reinstatement that occurred on timepoints when participants verbally recounted room details and on timepoints when they verbally described the objects that were paired to a particular room; in principle, room reinstatement during either set of timepoints could be acting to scaffold object retrieval. To this end, we computed two separate room reinstatement scores within the RRCN:

RRCN-room-recall: Room evidence extracted with the RRCN mask during timepoints in which participants were speaking about a room during free and guided recall.

RRCN-object-recall: Room evidence extracted within the RRCN mask during timepoints in which participants were speaking about the object that had been associated with a given room during guided and free recall.

To isolate the unique relationship between room reliability and object reinstatement, we regressed out both RRCN measures from each variable. Specifically, we fitted a linear model with ROCN object reinstatement as the dependent variable and both RRCN-room-recall and RRCN-object-recall as predictors. The ROCN residuals from this model represented object reinstatement variance unexplained by room reinstatement. Similarly, we fit a second linear model with room reliability as the dependent variable and the same two RRCN measures as predictors. The room reliability residuals from this model represented room reliability variance unexplained by room reinstatement. Finally, we computed a Pearson correlation between these two residuals. To test for significance, we ran a non-parametric permutation test in which we shuffled the ROCN residuals and recomputed the correlation 1,000 times to generate a null distribution of correlation values before running FDR correction for $q < 0.05$.

To identify regions where the relationship between room reliability and object reinstatement had a significant positive or negative change after controlling for room reinstatement, we ran a contrast in which the correlation values of the partial correlation were subtracted from the correlation values of our original model. To test for this difference, we computed a composite score of each correlation by averaging the results of each searchlight across recall task types (that is, guided and free recalls) and participants. Next, we computed the difference between the results of our original model and the partial correlation as well as on their permutations to get a null distribution of differences. To test for significance, we ran a non-parametric permutation test where we compared the true differences from the null distribution of differences and FDR-corrected for $q < 0.05$.

**Relationship between room reliability and room features.** Do properties of a room contribute to the reliability of their representation? We sought to identify whether physical or graph theoretical features

of a room contributed to the reliability of their representation. To do this, we used the 3D Unity model of the environment to compute a list of physical features such as total room volume, total volume occupied by background objects, the proportion of occupied volume and total room volume, area, object count, number of corners and whether the room has a window (that is, a view to the outside) and used the room adjacency matrix to compute graph-theoretical features such as degree, betweenness, closeness, eigenvector and pagerank. We then selected six features (degree, ratio of occupied volume, background object count, floor area, number of corners and 'has window') that were the least collinear and provided conceptually non-overlapping properties (for example, betweenness and degree are collinear). We z-scored each feature (except the binary 'has window') and then ran a searchlight analysis where we regressed room reliability on each of the six z-scored features for every participant. To test for statistical significance of each of the resulting beta coefficients, we ran a non-parametric permutation test where room reliability was shuffled 1,000 times within participants before regressing again on the features to generate a null distribution of beta coefficients. We then averaged across participants before running FDR correction on the resulting z values and thresholding at $q < 0.001$.

## Reporting summary

Further information on research design is available in the Nature Portfolio Reporting Summary linked to this article.

## Data availability

All data are openly available at https://openneuro.org/datasets/ds005704.

## Code availability

Scripts used for analysis are available via GitHub at https://github.com/rmasiso/MemoryPalaceReliability.

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

## Acknowledgements

We thank N. Keshavarzian, M. Nguyen, H. Hillman, S. Nastase, S. Zorowitz, T. Zalaback, S. Collin, J. Antony and everyone else for buddying during the long fMRI sessions; N. Cruz-Lebrón, J. Williams, the Norman lab and Baldassano lab members for insightful comments and feedback; S. Perrin, the VRTK and Unity community for valuable insight on optimizing VR gameplay; and J. Williams, J. Reske, N. Prillaman, J. Masís and Burne Holiday (J. Masís, J. Edelmann, C. Furlong and N. Tyrell) for providing beautiful music to accompany the rooms. We also thank J. Chen and C. Honey for their guidance and support. This work was supported by a Multi-University Research Initiative Grant (ONR/DoDN00014-17-1-2961) to K.A.N. and NINDS D-SPAN F99NS120644-01 and T32MH065214 to R.M.-O. The funders had no role in the study design, data collection and analysis, decision to publish or preparation of the manuscript.

## Author contributions

R.M.-O., K.A.N. and C.B. jointly conceptualized the study and designed the methodology. R.M.-O. curated and analysed the data, developed the experimental and analytical pipeline, created the visualizations and wrote the original draft. R.M.-O. and C.B. collected participant data. K.A.N. and C.B. provided guidance on analysis and interpretation, supervised the project and contributed to manuscript editing and revision.

## Competing interests

The authors declare no competing interests.

## Additional information

**Correspondence and requests for materials** should be addressed to Rolando Masís-Obando.

# Reporting Summary

## Statistics

For all statistical analyses, confirm that the following items are present in the figure legend, table legend, main text, or Methods section.

| n/a | Confirmed | |
|---|---|---|
| ☐ | ☒ | The exact sample size (*n*) for each experimental group/condition, given as a discrete number and unit of measurement |
| ☐ | ☒ | A statement on whether measurements were taken from distinct samples or whether the same sample was measured repeatedly |
| ☐ | ☒ | The statistical test(s) used AND whether they are one- or two-sided *Only common tests should be described solely by name; describe more complex techniques in the Methods section.* |
| ☐ | ☒ | A description of all covariates tested |
| ☐ | ☒ | A description of any assumptions or corrections, such as tests of normality and adjustment for multiple comparisons |
| ☐ | ☒ | A full description of the statistical parameters including central tendency (e.g. means) or other basic estimates (e.g. regression coefficient) AND variation (e.g. standard deviation) or associated estimates of uncertainty (e.g. confidence intervals) |
| ☐ | ☒ | For null hypothesis testing, the test statistic (e.g. *F*, *t*, *r*) with confidence intervals, effect sizes, degrees of freedom and *P* value noted *Give P values as exact values whenever suitable.* |
| ☒ | ☐ | For Bayesian analysis, information on the choice of priors and Markov chain Monte Carlo settings |
| ☐ | ☒ | For hierarchical and complex designs, identification of the appropriate level for tests and full reporting of outcomes |
| ☐ | ☒ | Estimates of effect sizes (e.g. Cohen's *d*, Pearson's *r*), indicating how they were calculated |

*Our web collection on statistics for biologists contains articles on many of the points above.*

## Software and code

Policy information about availability of computer code

| Data collection | Unity 3D (5.5.2f1 & 2017.1.2f1) for virtual environment creation and Virtual Reality Toolkit (VRTK), a virtual-reality programming tool-kit for Unity3D, for participant VR interactivity. PsychoPy for task presentation in the scanner, OBS Studio for video capture. Additional software used for 3D modeling included SketchUp and Blender. |
|---|---|
| Data analysis | Python 3.x (NumPy, SciPy, scikit-learn, MNE-Python), fMRIPrep v1.0.3 (Nipype-based), FreeSurfer v6.0.0, ANTs v2.1.0, FSL v5.0.9, AFNI v16.2.07. Visualization with Python and MNE-Python. Participant recalls were minimally transcribed using UPenn's TotalRecall. All analysis scripts are publicly available on GitHub: https://github.com/rmasiso/MemoryPalaceReliability -- custom algorithms (e.g., searchlight pattern reliability and reinstatement analyses) are described in the Methods and provided in the repository. |

For manuscripts utilizing custom algorithms or software that are central to the research but not yet described in published literature, software must be made available to editors and reviewers. We strongly encourage code deposition in a community repository (e.g. GitHub). See the Nature Portfolio guidelines for submitting code & software for further information.

## Data

Policy information about availability of data

All manuscripts must include a data availability statement. This statement should provide the following information, where applicable:

- Accession codes, unique identifiers, or web links for publicly available datasets
- A description of any restrictions on data availability
- For clinical datasets or third party data, please ensure that the statement adheres to our policy

> All raw and preprocessed MRI and behavioral data are openly available on OpenNeuro: https://openneuro.org/datasets/ds005704

## Research involving human participants, their data, or biological material

Policy information about studies with human participants or human data. See also policy information about sex, gender (identity/presentation), and sexual orientation and race, ethnicity and racism.

| | |
|---|---|
| Reporting on sex and gender | Both male and female participants were included in approximately equal numbers (16 females, 14 males initially; final analyzed sample: 11 females, 14 males).<br>Sex was recorded through participant self-report at enrollment and used solely for demographic description; no analyses were stratified or compared by sex or gender identity, as the study's hypotheses did not pertain to sex-related differences. |
| Reporting on race, ethnicity, or other socially relevant groupings | Participants were recruited from the Princeton University community and surrounding area.<br>Recruitment aimed to ensure that underrepresented minorities were included in approximate proportion to the local population.<br>Race or ethnicity were not collected as experimental variables and were not included in any analyses, as the study objectives concerned general principles of spatial memory and neural reinstatement. |
| Population characteristics | Healthy adults aged 21–32 years, with normal or corrected-to-normal vision, and no reported neurological or psychiatric disorders.<br>All participants were fluent in English and had prior experience using computers or gaming interfaces.<br>The final analyzed sample comprised 25 participants (11 female, 14 male). |
| Recruitment | Participants were recruited through the Princeton University research participation system, local flyers, and online postings. Eligibility criteria included normal or corrected-to-normal vision and no history of neurological or psychiatric illness. Recruitment aimed to balance gender representation and reflect the demographic composition of the local community. Participants provided written informed consent and received monetary compensation. |
| Ethics oversight | All procedures were approved by the Princeton University Institutional Review Board (IRB #7225).<br>All participants provided written informed consent prior to participation and were debriefed following the experiment.<br>The study complied with all relevant ethical regulations for research involving human participants. |

Note that full information on the approval of the study protocol must also be provided in the manuscript.

# Field-specific reporting

Please select the one below that is the best fit for your research. If you are not sure, read the appropriate sections before making your selection.

☐ Life sciences  ☒ Behavioural & social sciences  ☐ Ecological, evolutionary & environmental sciences

For a reference copy of the document with all sections, see nature.com/documents/nr-reporting-summary-flat.pdf

# Behavioural & social sciences study design

All studies must disclose on these points even when the disclosure is negative.

| | |
|---|---|
| Study description | This study investigated how reliable neural representations of spatial contexts (rooms) support memory reinstatement of associated objects. Participants learned a 23-room virtual environment in virtual reality (VR), after being scanned for neural snapshots of the rooms, they returned to VR to learn newly placed objects in each room, and then recalled the rooms and objects during fMRI scanning. The research combined immersive behavioral learning (in VR) with whole-brain fMRI analyses to link stable room representations ("spatial context reliability") to subsequent object reinstatement. |
| Research sample | Thirty healthy adults aged 21–32 years (16 female, 14 male) participated. All had normal or corrected-to-normal vision and no history of neurological or psychiatric conditions. Five participants were excluded due to missing or corrupted MRI data, leaving n = 25 for analysis. |
| Sampling strategy | Participants were recruited from the Princeton University community and surrounding area through flyers, mailing lists, and online postings. The target sample size was based on prior fMRI memory and reinstatement studies using similar within-subject designs (typically 20–30 participants), providing sufficient power for searchlight and cross-participant analyses. Recruitment aimed to balance |

| Data collection | gender representation and reflect local demographic diversity. |
|---|---|
| | Behavioral and VR data were collected in the Princeton Neuroscience Institute behavioral labs; MRI data were collected on a Siemens Prisma 3 T scanner with a 64-channel head coil.<br>Virtual reality tasks were implemented in Unity 3D using custom code and the Virtual Reality Toolkit (VRTK). Stimuli were presented in the scanner via PsychoPy. Speech recall data were recorded with the scanner's integrated microphone and annotated using TotalRecall. Data were preprocessed using fMRIPrep, FreeSurfer, FSL, AFNI, ANTs, and custom Python scripts. |
| Timing | Each participant completed the experiment across two consecutive days:<br><br>Day 1: Behavioral VR learning and map-drawing sessions (~1.5 hours).<br><br>Day 2: Two fMRI scanning sessions (pre-learning and post-learning, ~3 hours total) and an intermediate VR session (~0.5 hours).<br>Data collection occurred between 2018 and 2019 at Princeton University. |
| Data exclusions | Five participants were excluded due to missing or corrupted MRI files that prevented preprocessing and analysis.<br>No exclusions were made based on demographic characteristics. All other participants (n = 25) were included in every reported analysis. |
| Non-participation | All enrolled participants completed the full experimental protocol across both days, except those excluded due to technical data loss. No participant withdrew voluntarily. |
| Randomization | Room–object pairings were randomized independently for each participant to prevent systematic associations between specific rooms and objects.<br>The order of room and object videos during scanning was pseudorandomized within task blocks.<br>Classifier analyses used leave-one-participant-out cross-validation to eliminate circularity and ensure unbiased generalization across subjects. |

# Reporting for specific materials, systems and methods

We require information from authors about some types of materials, experimental systems and methods used in many studies. Here, indicate whether each material, system or method listed is relevant to your study. If you are not sure if a list item applies to your research, read the appropriate section before selecting a response.

## Materials & experimental systems

| n/a | Involved in the study |
|---|---|
| ☒ | Antibodies |
| ☒ | Eukaryotic cell lines |
| ☒ | Palaeontology and archaeology |
| ☒ | Animals and other organisms |
| ☒ | Clinical data |
| ☒ | Dual use research of concern |
| ☒ | Plants |

## Methods

| n/a | Involved in the study |
|---|---|
| ☒ | ChIP-seq |
| ☒ | Flow cytometry |
| ☐ | ☒ MRI-based neuroimaging |

## Plants

| Seed stocks | Plants were not used in this study. |
|---|---|
| Novel plant genotypes | Plants were not used in this study. |
| Authentication | Plants were not used in this study. |

## Magnetic resonance imaging

### Experimental design

| Design type | Within-subjects, cross-participant design. Participants completed both behavioral (VR) and fMRI tasks across two consecutive days. Day 1 focused on spatial learning in VR; Day 2 included pre- and post-learning fMRI scans as well as a short VR task in which participants had to learn the newly placed objects in the virtual environment rooms. |
|---|---|

| Design specifications | All participants underwent identical task sequences (VR environment learning, pre-learning scan, VR newly placed objects learning, and post-learning scan). |
|---|---|
| | Stimulus order within each scanning task was pseudorandomized. |
| | Classifier analyses used leave-one-participant-out cross-validation. |
| Behavioral performance measures | VR foraging scores, map-drawing accuracy, verbal recall duration, and recall contiguity were logged and analyzed for learning progression and memory retrieval structure. |

## Acquisition

| Imaging type(s) | Functional and structural MRI (BOLD EPI and T1-weighted anatomical imaging). |
|---|---|
| Field strength | 3 Tesla (Siemens Prisma scanner, 64-channel head coil). |
| Sequence & imaging parameters | Multiband EPI sequence (TR = 1300 ms, TE = 33 ms, flip angle = 80°) |
| | Slice thickness: 2 mm |
| | Field of view: 192 mm² |
| | Multiband factor (SMS) = 4 |
| | Whole-brain coverage |
| | T1-weighted anatomical: MPRAGE, 1 mm³ isotropic resolution |
| Area of acquisition | Whole brain, including cortical and subcortical regions. |

Diffusion MRI    ☐ Used    ☒ Not used

## Preprocessing

| Preprocessing software | fMRIPrep 1.0.3 (Nipype-based pipeline) |
|---|---|
| | FreeSurfer 6.0.0 for surface reconstruction |
| | ANTs 2.1.0, FSL 5.0.9, AFNI 16.2.07 for motion correction and registration |
| | Custom Python scripts (NumPy, SciPy, MNE-Python) for nuisance regression and z-scoring |
| Normalization | Surface- and volume-based normalization using ANTs nonlinear registration. |
| Normalization template | ICBM 152 Nonlinear Asymmetrical template (2009c) and fsaverage6 surface template. |
| Noise and artifact removal | Motion correction via MCFLIRT |
| | Slice-timing correction via AFNI 3dTshift |
| | "Fieldmap-less" distortion correction (Huntenburg et al., 2014) |
| | CompCor (aCompCor, tCompCor) noise regressors |
| | High-pass filtering (0.008 Hz cutoff) |
| | Regression of CSF, WM, and motion parameters |
| Volume censoring | No volume deletion; motion regressors were included as nuisance covariates. |

## Statistical modeling & inference

| Model type and settings | Voxel-wise GLMs for object and room regressors; searchlight-based multivariate analyses. |
|---|---|
| | Subsequent analyses used multivariate classification (multinomial logistic regression) and correlation-based reliability measures. |
| Effect(s) tested | Neural stability and distinctiveness of room representations ("room reliability") |
| | Relationship between pre-learning room reliability and post-learning object reinstatement |
| | Searchlight-based regressions predicting reliability from room features |

Specify type of analysis:    ☐ Whole brain    ☐ ROI-based    ☒ Both

| Anatomical location(s) | Our full hippocampus region of interest (ROI) was extracted from a freesurfer subcortical parcellation. This ROI was then split into an anterior portion (y > -20) and posterior portion (y <= -20) in MNI space (Guo et al., 2020; Poppenk et al., 2013; Masis-Obando et al., 2022). |

Statistic type for inference

(See Eklund et al. 2016)

| t-tests and Pearson correlations; nonparametric permutation tests for validation. |

Correction

| False Discovery Rate (FDR) correction applied at q < 0.05 (q < 0.001 for feature regressions). |

## Models & analysis

| n/a | Involved in the study |
| --- | --- |
| ☒ | ☐ Functional and/or effective connectivity |
| ☐ | ☒ Graph analysis |
| ☐ | ☒ Multivariate modeling or predictive analysis |

Graph analysis

| Conducted on room adjacency structure (degree, betweenness, closeness, pagerank) from VR environment |

Multivariate modeling and predictive analysis

| Multinomial logistic classifiers (leave-one-participant-out cross-validation) and correlation-based predictions linking room reliability to object reinstatement |

