## [Peer Review File · Nature Human Behaviour]

Spatial contexts with reliable neural representations support reinstatement of subsequently placed objects

Corresponding Author: Dr Rolando Masís-Obando

Version 0:

Decision Letter:

4th February 2025

Dear Dr Masís-Obando,

Thank you once again for your manuscript, entitled "How sturdy is your memory palace? Reliable room representations predict subsequent reinstatement of placed objects", and for your patience during the peer review process.

Your Article has now been evaluated by 3 referees. You will see from their comments copied below that, although they find your work of potential interest, they have raised quite substantial concerns. In light of these comments, we cannot accept the manuscript for publication, but would be interested in considering a revised version if you are willing and able to fully address reviewer and editorial concerns.

We hope you will find the referees' comments useful as you decide how to proceed. If you wish to submit a substantially revised manuscript, please bear in mind that we will be reluctant to approach the referees again in the absence of major revisions. We are committed to providing a fair and constructive peer-review process. Do not hesitate to contact us if there are specific requests from the reviewers that you believe are technically impossible or unlikely to yield a meaningful outcome.

In particular, Reviewer #1 expresses concerns regarding the evidence in support of the scaffolding mechanism. They also remark that it is currently unclear whether neural representations reflect rooms or objects. Finally, all reviewers ask for a more thorough examination of the behavioral data and the features of the rooms or objects that may be driving the observed effects. We ask that you address these concerns in full. Although we have no doubt that reviewer suggestions to show generalization of the results to other mnemonic methods or emotional states would strengthen your manuscript, we believe that such work is best left as a future research project.

If you wish to submit a suitably revised manuscript, we would hope to receive it within 4 months. I would be grateful if you could contact us as soon as possible if you foresee difficulties with meeting this target resubmission date.

- Include a "Response to the editors and reviewers" document detailing, point-by-point, how you addressed each editor and referee comment. If no action was taken to address a point, you must provide a compelling argument. When formatting this document, please respond to each reviewer comment individually, including the full text of the reviewer comment verbatim followed by your response to the individual point. This response will be used by the editors to evaluate your revision and sent back to the reviewers along with the revised manuscript.
- Highlight all changes made to your manuscript or provide us with a version that tracks changes.

Link Redacted

Thank you for the opportunity to review your work. Please do not hesitate to contact me if you have any questions or would like to discuss the required revisions further.

Sincerely,

Nature Human Behaviour

Reviewer expertise:

Reviewer #1: Episodic memory, fMRI, MVPA

Reviewer #2: Episodic memory, fMRI, MVPA

Reviewer #3: Episodic memory, fMRI, MVPA

REVIEWER COMMENTS:

Reviewer #1 (Remarks to the Author):

In this study, Masalis-Obando and colleagues developed a virtual reality (VR) "memory palace" to explore the hypothesis that spatial locations characterized by distinctive and stable neural representations can effectively support the encoding and reliable retrieval of new episodic memories. Their findings indicated that a participant-specific room reliability measure could predict object reinstatement throughout the cortex.

Overall, the study tackles an intriguing aspect of memory research. However, in its current form, I have some reservations about the findings and feel that the study provides limited mechanistic insights.

Firstly, I have doubts whether the experimental design is optimally suited to address the research question. The authors aimed to assess if the reliability of context, specifically room representation, could facilitate object reinstatement. In the recall task, participants were asked to perform several verbal recall tasks. For both guided and free recall tasks, participants needed to recall the rooms and the objects, without a clear rationale for choosing these task types. Additionally, this design presents challenges in distinguishing the neural representations of rooms from those of objects, potentially leading to inaccuracies in the object template based on retrieval-related activation in the retrieved object classifier network (ROCN). While a perceived object classifier network (POCN) was included, it did not overlap with the ROCN. Moreover, if participants struggled to recall the rooms accurately, it might impact their ability to recall associated objects, particularly in the free recall tasks.

Secondly, the authors could enhance the study by providing a more detailed analysis of behavioral data to reflect the quality of object memory. Unfortunately, the behavioral data analysis appears brief, with only one sentence mentioning behavioral results: "Across both recall types, participants' recollections were at ceiling, with 92% and 76% of participants achieving over 90% recall accuracy for guided and free recalls, respectively." Clarification on what constituted a correct response would be helpful since participants were asked to describe room and object details, and yet, these recalled details were not reported. Additionally, 76% of participants scoring over 90% might not be categorized as a ceiling performance.

Thirdly, further elaboration on the presentation of results would be beneficial. Beyond the numbers provided for behavioral performance, the results section lacks comprehensive data presentation. It would be helpful to include statistics, bar graphs, scatter plots, and individual participant data. Information on room representation reliability and reinstatement scores, as well as the strength of their correlation, would greatly aid in evaluating data quality.

Fourthly, in the context of participant-specific versus group-averaged results, regions identified with superior prediction using the original model were discussed. However, it remains unclear whether "better" corresponds to statistically or numerically greater predictions, and clarification on these differences would be valuable.

Lastly, the study offers limited mechanistic explanations for the observed correlations. For instance, evidence supporting scaffolding is lacking. The method of loci (MOL) emphasized in this study involves learning the sequence of rooms. It would be insightful to explore how the learning of sequence order influences the reliability and distinctiveness of rooms and object recall. Furthermore, while examining whether this correlation applies to any paired association task or is specific to the method of loci (MOL) might exceed the current scope, discussing this aspect could offer valuable insights.

Reviewer #2 (Remarks to the Author):

In their paper "How sturdy is your memory palace? Reliable room representations predict subsequent reinstatement of placed objects" the authors explore a classic question in the field of memory – why are spatial contexts a scaffold for episodic memory? Specifically, they test the prediction that the more reliable and distinct a spatial context representation, the better item encoding and subsequent recall will be in that context. Evidence of item recall here is at the neural, not behavioral level. To test this, they designed a 23 room virtual reality space. Participants first explored the space, completing a foraging task. The next day they returned and were shown videos of the rooms in the scanner, room reliability was calculated as similarity of a room's representation to itself (across runs) relative to its similarity with the representation of other rooms. After obtaining this reliability metric, participants explored the same space with new objects placed throughout the rooms and were instructed to memorize the objects. On the same day, following the learning period, they went back into the scanner and completed a series of recall tasks. Item reinstatement was measured by training a classifier in a separate cued-recall task. Researchers found a significant positive correlation between room reliability and item reinstatement, supporting their hypothesis that the more distinct and reliable a spatial context, the more richly detailed the subsequent memory would be.

This is an ambitious, well-designed, and well-executed study. The methods are solid and rigorous, figures are clear, and the manuscript is extremely well-written. We have some minor comments that may add to the manuscript, following minor revisions we recommend it for publication.

Comments

-The authors could cover a little more literature on the timing of spatial context reinstatement vs. object reinstatement, this could help bolster the argument in the introduction that spatial context is a scaffold/special

-It might help to define spatial context, both broadly when discussing the background literature, and how it is defined experimentally here. If we understood correctly, in this paradigm it is a room with objects and not an empty room (again something we didn't realize until we got to the methods) and there is music/sounds distinct to that context. This spatial context is then populated with and more prominent (random?) objects for the memory encoding. This feels like a more rich/naturalistic definition of context, but is it similar to what others have done in the past? Does that change anything?

-It wasn't clear to us why they did two different types of recall? (though a strength to find the same results across both)

-Anything special in terms of features about the rooms that were most reliable across participants? How much individual variability?

-Could you do the same study/analysis for emotional states? Would you expect the same results?

-A discussion on the relationship between familiarity/novelty and context, and how that maps on to reliability/distinctiveness/memory benefits could be helpful. Is the novelty effect in memory separate from spatial context? Or do we get a benefit because novel spatial contexts are distinct (even if not reliable) on account of their novelty? Or does it need to be novel and dissimilar in experience (Asian city block vs. different US city block for a person from the US who has never been to Asia). How do repeated spatial contexts stay distinct and useful? Any plans to have more long-term familiarity tests?

Very Minor Comments

-Auditory cortex is one of the most reliable brain areas for room reliability representations. This seems surprising, until you read on in the methods and find out they played distinct music/soundtracks for the room. Might want to pop that in earlier.

-line 57 "may enhance" does it or not? Is the literature mixed?

-Reviewed by Anna Blumenthal & Jordan DeKraker

Reviewer #3 (Remarks to the Author):

This paper uses a clever combined VR-fMRI approach to create novel VR memory palaces for participants, and then test how neural stability of representations for the palace rooms relates to ability to reinstate memories for objects within those rooms. Overall, I found this work incredibly compelling, interesting, and well-executed. This work shows a fantastic way to intentionally design memory palaces, observe a neural measure of their effectiveness, and it shows a direct connection to neural measures of memory for associated items. The writing is also very clear. I only have relatively minor comments about the work that are largely just curiosities that emerge.

- What do you think is the cognitive mechanism behind the link of room reliability and object reinstatement? Is it that more reliable rooms serve as better cues to an association with the objects?

- In the main text, you discuss participant-specific reinstatement regions, but in the Methods you note the importance of considering the group-wise regions too--they have different implications for how to think about what makes a good memory palace. Currently it's hard to sense the extent of the group-wise effect because right now the group-wise effect is using N-1 participants (vs. 1 participant for the participant-wise effect), so by nature of having more participants it may show stronger effects. I wonder if there's an analysis that could show what the specific group-wise effect would be -- for example by iteratively using one participant's room reliability to predict another participant's object classifier evidence, averaged across many iterations. (So the participant-specific and group-wise measures both use only one participant's data per task at a time).

- Since a part of this study's central claim is that some rooms create more robust representations than others, I'd be curious to know--what were the rooms that were worse/better? Were there any features that might relate to their reliability, like room size, number of connecting rooms, or number of background objects? (The latter is helpful to test because it is posed in the discussion, too). Were there any objects that also tended to have more evidence over others across rooms? (Though this is not as central a part of the study--am just curious if it goes both ways).

- Reading the caption of Figure 3A, it sounds like the classifier was trained on N-1 participants viewing the object to be classified, while the text makes it sound like it was trained on N-1 participants retrieving the object from memory when in its associated room. Which is it (perception or memory), or am I misinterpreting the analysis? (I do see you have a POCN too but this Figure is specifically labeled ROCN).

- In the paragraph starting on line 751 (Section 5.8.5), did this measure of object evidence only use trials where participants correctly recalled the object (e.g., "teddy bear"), or did it also include false recollections? It might help to have a little text justifying the decision since I can see why one would do it either way, but it does slightly change the interpretation (i.e., looking at correct reinstatement, or looking at any time a participant reports recalling an item even if it wasn't really there).

Version 1:

Decision Letter:

Our ref: NATHUMBEHAV-24125027A

29th September 2025

Dear Dr Masís-Obando,

Thank you for submitting your revised manuscript "How sturdy is your memory palace? Reliable room representations predict subsequent reinstatement of placed objects" (NATHUMBEHAV-24125027A). It has now been seen by the original referees and their comments are below. As you can see, the reviewers find that the paper has improved in revision. We will therefore be happy in principle to publish it in Nature Human Behaviour, pending minor revisions to comply with our editorial and formatting guidelines.

We are now performing detailed checks on your paper and will send you a checklist detailing our editorial and formatting requirements within two weeks. Please do not upload the final materials and make any revisions until you receive this additional information from us.

Sincerely,

Nature Human Behaviour

Reviewer #1 (Remarks to the Author):

The authors have done a great job addressing all my comments.

Reviewer #2 (Remarks to the Author):

The revised paper is significantly improved, and the authors more than adequately responded to all of my points. I am happy

to recommend this paper for publication!

Reviewer #3 (Remarks to the Author):

The authors have done a fantastic job at addressing my comments. Their added text and figures have helped to make the methods much clearer. I also appreciate the new analyses looking at room features, the group-wise analysis that now uses one participant at a time for training the classifier, and the new analysis partialing out the RCCN. Overall this is exciting, innovative, and groundbreaking work on how memories are formed and bound to spatial locations.

Dear Editor and Reviewers,

We thank you for the thoughtful and constructive feedback on our manuscript, "How sturdy is your memory palace? Reliable room representations predict subsequent reinstatement of placed objects". The comments provided by each of you have been invaluable. We have carefully considered all points raised and have revised the manuscript accordingly. These revisions have significantly improved the clarity, rigor and impact of our work. We are excited about these changes and the new and improved manuscript – so, thank you very much for your time, comments, and consideration.

Below, we provide an overview of major changes that we've made, followed by a point-by-point response to each reviewer's comments. The manuscript changes have been color-coded in blue below each reviewer response.

Overview of Major Changes

Across all three reviews, several recurring themes emerged. First, reviewers asked for a clearer definition and operationalization of "scaffolding," specifically, clarification on the mechanism by which room reliability supports subsequent memory. Second, reviewers requested clarification on the rationale behind including both guided and free recall tasks. Third, reviewers highlighted the need for expanded data presentation in the neural and behavioral analyses. Fourth, reviewers raised concerns about the potential confound between room and object representations. Finally, reviewers requested a fairer approach to our participant-specific versus group-level model comparison.

Regarding scaffolding, our primary hypothesis was that reliable room representations scaffold memory for objects by facilitating the binding of objects to rooms at encoding. Our finding that room reliability (measured prior to encoding) correlates with object reinstatement (measured during recall) is compatible with this "facilitated binding at encoding" hypothesis. However, the reviewers suggested an alternative explanation for this result – namely, that room reliability (measured prior to encoding) could promote successful room reinstatement during recall periods, which then promotes object reinstatement; this could give rise to a correlation between room reliability and object reinstatement, even in the absence of facilitated binding at encoding. To address this alternative hypothesis, we added a new analysis in which we statistically controlled for room reinstatement during verbal recall. We found that the relationship between room reliability and object reinstatement remained significant, indicating that room reinstatement (at recall) can not be the sole factor driving this effect, and thereby providing indirect evidence for our primary hypothesis (facilitated binding at encoding). We enriched the Introduction to set up this distinction, added a new figure and analysis (partial correlation; new Figure 6C), and updated the Methods and Discussion accordingly.

Several reviewers also questioned the rationale for including both guided and free recall tasks and why their results were originally presented separately. Upon revisiting these analyses, we discovered and corrected a minor indexing error in the computations linking room reliability with object reinstatement. This error only affected those correlational analyses, but once fixed, the resulting surface maps (originally Figure 5A and 5B; now Figure 6A - Supplement 2) became far more consistent, symmetric, and robust, while preserving the main effects (positive relationships in precuneus, parietal cortex, and dorsal occipital cortex; negative relationships in parahippocampal cortex and auditory cortex). The correction also revealed additional regions of positive association shared across both tasks, including

superior frontal gyrus and insula. Given these improvements, the strong convergence across the two recall types, and reviewer concerns about the split presentation, we now statistically combine guided and free recall results into a single composite map as our primary result (see Figure 6 for ROCN and Figure 6 - Supplement 1 for POCN), while retaining the separated results for reference (Figure 6 – Supplement 2).

In response to multiple comments requesting deeper exploration of the data, we have added four new figures. Figure 3 now visualizes our behavioral scoring approach and accuracy distributions for both guided and free recall (which we now hope shed light on the behavioral ceiling effects). Figure 3 – Supplement 1 shows time spent recalling rooms and objects, along with an analysis of contiguous mental traversal during free recall. Figure 6 – Supplement 3 presents subject-level scatterplots to illustrate the robustness of the room reliability–object reinstatement relationship in an example searchlight. Finally, Figure 2 – Supplement 1 examines how six physical and structural room features (e.g., size, number of objects, connectedness) relate to room reliability across the brain. Corresponding updates have been made throughout the Results, Methods, and Discussion.

With respect to concerns about potential confounds between room and object representations, we have revised the Methods to clarify that our analyses were specifically designed to decorrelate room and object classification. We now explicitly explain why our analyses are not circular, why our room templates (and therefore our room reliability measure) are free from object information, and how our object classifiers, which were trained on N-1 participants' object templates, were also free of room-related information for the held-out participant due to room-object pairings being randomized across participants. In addition, our new partial correlation analysis shows that the observed relationship between room reliability (at encoding) and object reinstatement (at retrieval) persists when we control for room reinstatement at retrieval.

Lastly, based on reviewer feedback, we have changed our model comparison approach. Instead of comparing the participant-specific model to an averaged N-1 group model (which could overweight the group effect), we now iteratively compare each participant's model to every other participant's model individually, yielding a fairer and more balanced comparison (Figure 6B; 6B-Supplement 1; 6B and 6D-Supplement 2). We have also made a few cosmetic changes to other figures to remove clutter and ensure clarity.

Overall, we have made substantial changes to the main text by heavily revising the Introduction, Results, Discussion and Methods, including the addition of 8 new figures. Again, we are grateful for the reviewers' and editor's feedback, which has led to a substantially improved and more rigorous manuscript.

Response to Reviewer #1

We would like to thank you for your detailed assessment and insightful comments which have greatly improved our manuscript! We also appreciate your recognition of our study tackling an intriguing aspect of memory research. Please find below your comments and how we addressed them.

Comment 1: Concerns over the rationale for the recall tasks

Comment:

“Firstly, I have doubts whether the experimental design is optimally suited to address the research question. The authors aimed to assess if the reliability of context, specifically room representation, could facilitate object reinstatement. In the recall task, participants were asked to perform several verbal recall tasks. For both guided and free recall tasks, participants needed to recall the rooms and the objects, without a clear rationale for choosing these task types.”

Response:

Thank you for this thoughtful comment. We appreciate the opportunity to clarify the rationale behind our recall tasks.

First, we designed the guided and free recall tasks to complement each other in assessing memory in two different and ecologically valid ways. Unlike the guided recall tasks, in which participants were presented with the names of the 5 rooms they had to recall, free recall tasks involved uncued retrieval of room and object details. Participants were not instructed in how to recall, and the recalls for both tasks were temporally unstructured and self-paced. These two tasks thus, in theory, would shed light into ecologically valid real-world episodic memory retrieval, since realistic verbal recall can sometimes be prompted with cues and sometimes it can be fully self-initiated.

Overall, given the confusion about presenting the results of these two tasks separately, a similar concern raised by Reviewer #2, and the similarity of our updated results between Guided and Free Recalls (which provided an internal replication), we have statistically combined these results into a single figure (now Figure 6A and 6B; formally Figure 5A/C and B/D). Put simply, we created a composite map where we averaged the results of Figure 5A/C with Figure 5B/D into a new set of brainmaps (Figure 6A and 6B). We tested for significance by running permutation tests on their combined null and ran FDR-correction on the resulting p-values. Because of your feedback, we have made substantial revisions to the Results and Methods section to account for the new composite map – and we also summarize it briefly in the Discussion when interpreting the robustness of the effects across task types. For the curious reader, we have kept the separate results of guided and free recalls in a new supplementary figure (Figure 6 - Supplement 2). To see the updated composite map of room reliability and ROCN object evidence as well as the complementary POCN analysis, please refer to our response to Reviewer #1 - Comment 2 for the new figures: Figure 6 and Figure 6 - Supplement 1, respectively.

We also made minor adjustments throughout the manuscript and replaced some of the sentences that alluded to both guided and free recalls with “verbal recall” or just “recall” to enhance flow and clarity about our results. To avoid cluttering our response in this document, these minor changes can be found color coded in blue in the manuscript.

Lastly, we note that our self-paced recall tasks ask participants to report both objects and rooms. It’s possible that different results would be obtained if we only asked participants to report objects. Future studies using alternative recall tasks (unrelated to room features) could shed light on this possibility.

Revised Results 1 (starting at line 243):

...We then averaged these correlations across participants to obtain a searchlight map that we then statistically averaged across recall task types (i.e., guided and free recalls) to get a composite map that indicated regions where room reliability in those regions correlated with subsequent object reinstatement (throughout the ROCN network; Fig 6A). Notable positive relationships were observed throughout parietal cortex, prefrontal cortex, superior frontal gyrus, insula, and precuneus. We also found notable negative relationships in right parahippocampal cortex, parts of the motor system, auditory cortex, and ventral visual regions. Importantly, when looking at this relationship separately for guided and free recalls (before generating our composite map), the regions revealed were highly similar, providing an internal replication of this relationship across two categorically different recall task types (see Figure 6 - Supplement 2)

Revised Results 2 (starting at line 289):

...In a similar fashion to how we related room reliability with object evidence within the Retrieved Object Classifier Network (ROCN), we ran a supplementary analysis in which we quantified object reinstatement within the Perceived Object Classifier Network (POCN; largely composed of visual regions) during verbal recall (Fig 6 - Supp 1). Across participants, we found generally similar results to the ROCN results, with a positive relationship between POCN reinstatement evidence and room reliability in parietal cortex, superior frontal gyrus, insula, posterior medial cortex, and dorsal occipital cortex.

Revised Methods 1 (starting at line 934):

5.8.6 Relationship between room reliability and object reinstatement: We hypothesized that rooms with more reliable representations in the pre-learning scans would be associated with higher levels of object reinstatement during self-paced verbal recall. To do this, we ran a searchlight analysis where we correlated the reliability of a room (see **Room reliability** methods section) with the network's evidence for the object paired to that room (see **Object classifier network selection** methods section). We did this for every room-object pair within a participant. For example, for a particular participant, the 23 room reliabilities were correlated with the corresponding 23 object reinstatement probabilities from the retrieval network. Afterwards, we averaged the Fisher-z transformed correlations across participants and recall task types (i.e., guided vs free recall) to generate a single composite correlation map. To test for statistical significance, we ran a nonparametric permutation test in which we randomly shuffled the object labels 1000 times to generate a null distribution of correlations within participants and for both recall types. Significance testing was then performed using the combined null distribution, and resulting p-values were FDR-corrected (see Figure 6). For reference, the results for each recall task type individually are presented in Figure 6 - Supplement 2.

Revised Methods 2 (starting on line 981):

5.8.7 Benefit of participant-specific room reliability: ... (and thus the observed results can not be entirely due to the group-wise effect). To test for statistical significance, we ran a nonparametric permutation test where the object labels were randomly shuffled 1000 times to generate a null distribution of model performance for each model. To generate a single composite map summarizing model performance across recall tasks, we averaged R^2 values across guided and free recalls for each model separately, computed the difference in R^2 , and tested for significance using a nonparametric permutation test on the combined null distribution

of R^2 differences, followed by FDR-correction on the resulting p-values. For completeness, separate results for guided and free recall are provided in Figure 6 – Supplement 2, while the main results reflect the combined composite analysis (see Figure 6).

New Figure 6 - Supplement 2

Figure 6 - Supplement 2: Predicting ROCN and POCN object reinstatement from room reliability for guided and free recalls. Relationships between room reliability and classifier network object reinstatement evidence. ROCN and POCN results are shown to the left and right of the dividing gray line, respectively. (A.) Regions where room reliability predicted ROCN object reinstatement in guided recalls (GR; left) and free recalls (FR; right). (B.) Model comparison results. Regions where room reliability predicted ROCN object reinstatement and there was a predictive benefit from participant-specific room reliability, shown separately for guided and free recalls. (C.) Regions where room reliability predicted POCN object reinstatement in guided recalls (GR; left) and free recalls (FR; right). (D.) Model comparison results. Regions where room reliability predicted POCN object reinstatement and there was a predictive benefit from participant-specific room reliability, shown separately for guided and free recalls. The surface maps presented in A and C were statistically thresholded by comparing correlations to a null distribution and then FDR-correcting for $q < 0.05$. The surface maps presented in B and D show the intersection of the participant-specific models (A and C) and the regions where there was a significant positive difference in the coefficient of determination between the original participant-specific model and the N-1 group model. Statistical significance for the differences between the coefficients of determination was determined by comparing the differences to a null distribution and FDR-correcting for $q < 0.05$. All surface maps are colored based on the magnitude of the z-scored correlation values of the participant-specific model, with blue showing negative and red showing positive relationships, respectively.

Revised Discussion (starting on line 447):

However, in our paradigm, participants were explicitly using a context-based strategy for retrieving items, mentally simulating rooms and trajectories through rooms in order to reinstate item memories. In this case, we would expect that having a reliable contextual index for episodic memories would be critical for effective recall of items, consistent with our findings that stability in scene-related brain regions predicted item reinstatement. Future work could investigate whether this relationship disappears or reverses in other situations, such as **when** many items are paired with the same room (reducing the usefulness of rooms as memory cues), **when** rooms **have** features that **vary**, e.g., with time of day (such that representational variability might reflect meaningful changes in contextual features), **or when the recall task requires reporting only objects while** suppressing recall of room features. Similarly, novelty may influence how room reliability scaffolds memory: a new context may be less stable than a highly-familiar location, but could still enhance memory because its novelty promotes additional attention and processing. Future work examining how repeated exposure and contextual novelty interact with reliability could shed new light on their contributions to memory.

Comment 2: Potential confounding of room and object representations

Comment:

“Additionally, this design presents challenges in distinguishing the neural representations of rooms from those of objects, potentially leading to inaccuracies in the object template based on retrieval-related activation in the retrieved object classifier network (ROCN).”

Response:

Thank you for bringing these points to our attention. While we can understand the concern about potentially overlapping neural signals for rooms and objects, we took several steps to mitigate these potential issues through our experimental design and analyses. Room and object representations were not confounded because:

1) Room templates used to calculate room reliability were computed using BOLD data acquired before any room-object pairings were learned: Because we were interested in how room reliability before the encoding of an event was predictive of the memory of that event, we computed room reliability using spatial context representations that were uncontaminated by the objects that were encountered later in the experiment. In other words, our room templates and therefore our room reliability measure are not conflated with subsequent object information.

2) Room-object pairings were randomized across participants: The room-object pairings were randomized for every participant, and the object templates were created by averaging object patterns across N-1 participants. Although the object templates for each specific participant in this N-1 group may contain information about the rooms associated with these objects for that participant, these associations cannot drive accurate object decoding in the left-out participant since their room-object pairings will be unrelated to the other participants' room-object pairings.

We have now revised the Methods to ensure these points are described more clearly.

Revised Methods 1 (starting at line 790):

5.8.3 Room reliability: We hypothesized that, in order for a room to serve as an effective retrieval cue for associated memories (i.e., objects paired to rooms), the neural representation for that room must be stable over time and distinct from other room patterns. We captured these properties with a composite measure we called room reliability. Crucially, this measure was computed based on data that were collected prior to participants learning the room-object associations. This ensured that our room templates, and therefore our room reliability measure, were not confounded with object information.

Revised Methods 2 (starting at line 836):

...The fact that each of the 24 participants in the training dataset had their own set of random room-object pairings ensured that the classifier was able to learn object representations that were not contaminated with room information (by contrast, if we had used a within-participant classification approach, room and object information would have been confounded, since objects were only scanned after they had been paired with a particular room). In other words, because room-object pairings were randomized for every participant, and object evidence for each participant was classified based on object templates derived from the other N-1 participants, any room-related information in the object templates would be unrelated to the room-related information in this left-out participant.

Comment:

“While a perceived object classifier network (POCN) was included, it did not overlap with the ROCN.”

Response:

We also understand your confusion regarding ROCN's and POCN's lack of overlap. First, we should again clarify that the differences between these networks are not due to room-related contamination for the ROCN; both of these networks use the procedure described above to define templates from the other N-1 participants (who have different room-object pairings), ensuring that room information cannot drive object decoding in the test participant. The only difference between ROCN and POCN is when we applied the classifier. For ROCN, we asked the question: where can we decode objects when participants are watching videos of *rooms*? For POCN, we asked: where can we decode objects when participants are watching videos of *objects*? So, while for ROCN we tested for object retrieval, for POCN we tested for object perception. Given the well-established neural differences between perception and retrieval, we expected to see differences in the best performing searchlights for these two classifiers, and indeed found that POCN was focused on sensory regions while ROCN was distributed across higher-level cortical regions. Because the ROCN was based on a task (verbal retrieval of objects) that was most similar to our self-paced verbal recall tasks, we used object evidence in the ROCN as our primary measure of object reinstatement during recall. For completeness, and because it is possible regions involved in the perception of objects may also be involved during unpaced verbal recall, we also used POCN in downstream analyses and illustrate results in our supplementary figures.

We have revised the Results section when we describe ROCN (and POCN briefly) to highlight these points.

Revised Results 2 (starting on line 215):

2.4 Retrieved Object Classifier Network (ROCN): To measure evidence of object reinstatement during self-paced (guided and free) verbal recall, we first needed to identify a network of regions that represent information about specific objects that were retrieved from memory; to select these regions in a non-circular fashion, we defined these regions using data from room-video object recall trials (Fig 4A). In these trials, participants viewed videos of all rooms and verbally recalled the object that had been assigned to each room as it was presented (Fig 4A). We used a leave-one-participant-out cross-validation procedure, whereby we made a neural template for each object based on object videos from N-1 participants, and then we used these templates to classify the (not-visibly-present) objects being recalled during room viewing in the held-out participant (Fig 4A). We opted for this across-participants approach (rather than classifying within-participants) because objects and rooms are confounded within participants, so room information could “leak” into training of a within-participant object classifier; this confound does not exist if training and testing are done across participants, each of whom has their own random set of room-object pairings. **In other words, the left-out participant’s object templates were never used to classify their own object recall during room videos.** We used this procedure to identify the top 50 best object classifier searchlights (~3% of all searchlights) to make our Retrieved Object Classifier Network (ROCN; Fig 4B), which we used as a mask (Fig 4D) when measuring object reinstatement evidence during the guided and free recall tasks. We found that the top classifier searchlights were spread throughout cortex and included regions in anterior temporal cortex, frontal gyrus, posterior temporal cortex, posterior medial cortex, and superior parietal cortex, among others (Fig 4C and Fig 4D). We also conducted **additional analyses to extract two other networks: For one, we classified object patterns while participants watched videos of objects (rather than retrieving object memories) to extract the Perceived Object Classifier Network (POCN), which was entirely, and unsurprisingly, due to the visual task, concentrated in early visual cortex (Fig 4 - Supp 1). For the other, we classified room patterns while participants watched videos of objects (analogous to ROCN, which classified object memories during room videos) to extract the Retrieved Room Classifier Network (RRCN), which was widely distributed and included precuneus, medial prefrontal cortex, anterior temporal cortex and visual cortex (Fig 4 - Supp 2).**

Comment:

“Moreover, if participants struggled to recall the rooms accurately, it might impact their ability to recall associated objects, particularly in the free recall tasks.”

Response:

Lastly, we also understand your concern that if participants struggle to remember rooms, they may struggle to remember objects too. In other words, functionally, object activation may be related to room activation at retrieval. To control for this possibility, we conducted a new partial correlation analysis where we regressed room reinstatement scores (computed during recall periods, using a network mask specialized for room reinstatement: the Retrieved Room Classifier Network or RRCN) from both ROCN object reinstatement and pre-learning room reliability, and then correlated the residuals. We found that, even after controlling for room reinstatement during recall, room reliability continued to significantly predict object reinstatement in ROCN (new Figure 6). To statistically determine whether this new partial

correlation revealed a change in the size of the relationship between room reliability and object reinstatement, we computed the difference between the original and partial correlation results across participants and recall task-types and tested for significance. Even after FDR correction, the contrast revealed no significant changes.

To accommodate these new results, we have made substantial adjustments to the manuscript. We have included a new Methods section describing how we extracted the RRCN mask (including a new Figure 4 - Supplement 2 to illustrate) and how we computed our new partial correlation analysis. We have also updated the Results section to reference our new RRCN mask and to report the partial correlation results. Finally, we have also updated the Discussion section to account for these new findings. We are very grateful for your comments as our revisions have greatly enhanced the rigor of our manuscript.

New Figure 6

A. Predicting object reinstatement from room reliability

B. Participant-specific benefit when predicting object reinstatement from room reliability

C. Controlling for room reinstatement

Predicting object reinstatement from room reliability (controlling for room reinstatement)

Figure 6: Predicting ROCN object reinstatement from room reliability. (A) Relationship between ROCN object reinstatement and room reliability. Regions where room reliability predicted ROCN object reinstatement across both guided and free recalls. Objects placed in rooms with the most pre-learning neural stability in these regions were reinstated the most strongly during retrieval. (B) Model comparison results. Regions where room reliability predicted ROCN object reinstatement across both guided and free recalls and there was a predictive benefit from participant-specific room reliability. In these regions, the rooms that were most reliable for a specific participant (rather than rooms that were generally reliable across the group) were predictive of object recall for that specific participant. The surface maps presented in B show the intersection of the participant-specific models shown in A and the regions where there was a significant positive difference in the coefficient of determination between the original

participant-specific model and the N-1 group model. Statistical significance for the differences between the coefficients of determination was determined by comparing the differences to a null distribution and FDR-correcting for $q < 0.05$. (C) Controlling for room reinstatement. Left column: Schematic illustrating how room reinstatement evidence in RRCN (during timepoints in which participants verbally recalled a room or its paired-object) was regressed out of room reliability and ROCN object reinstatement scores. Room reliability residuals were then correlated at each searchlight with ROCN object reinstatement residuals. Right column: Regions where room reliability predicted ROCN object reinstatement after controlling for room reinstatement during room and object recall. The surface maps presented in A and C were statistically thresholded by comparing correlations to a null distribution and then FDR-correcting for $q < 0.05$. All three surface maps are colored based on the magnitude of the z-scored correlation values of the participant-specific model, with blue showing negative and red showing positive relationships, respectively.

New Figure 6 - Supplement 1:

A. Predicting object reinstatement from room reliability

B. Participant-specific benefit when predicting object reinstatement from room reliability

C. Controlling for room reinstatement

Predicting object reinstatement from room reliability (controlling for room reinstatement)

Figure 6 - Supplement 1: Predicting POCN object reinstatement from room reliability. (A) Relationship between POCN object reinstatement and room reliability. Regions where room reliability predicted POCN object reinstatement across both guided and free recalls. Objects placed in rooms with the most pre-learning neural stability in these regions were reinstated the most strongly during retrieval. (B) Model comparison results. Regions where room reliability predicted POCN object reinstatement across both guided and free recalls and there was a predictive benefit from participant-specific room reliability. In these regions, the rooms that were most reliable for a specific participant (rather than rooms that were generally reliable across the group) were predictive of object recall for that specific participant. The surface maps presented in B show the intersection of the participant-specific models shown in A and the regions where there was a significant positive difference in the coefficient of determination between the original

participant-specific model and the N-1 group model. Statistical significance for the differences between the coefficients of determination was determined by comparing the differences to a null distribution and FDR-correcting for $q < 0.05$. (C) Controlling for room reinstatement. Left column: Schematic illustrating how room reinstatement evidence in RRCN (during timepoints in which participants verbally recalled a room or its paired-object) was regressed out of room reliability and POCN object reinstatement scores. Room reliability residuals were then correlated at each searchlight with POCN object reinstatement residuals. Right column: Regions where room reliability predicted POCN object reinstatement after controlling for room reinstatement during room and object recall. The surface maps presented in A and C were statistically thresholded by comparing correlations to a null distribution and then FDR-correcting for $q < 0.05$. All three surface maps are colored based on the magnitude of the z-scored correlation values of the participant-specific model, with blue showing negative and red showing positive relationships, respectively.

New Methods for RRCN Mask (starting at line 889)

3c) Retrieved Room Classifier Network Selection: We applied a similar procedure described in the **Retrieved Object Classifier Network selection** section, but instead of classifying non-visible object identity from post-learning room videos, we classified room identity from the post-learning object videos where objects (*but not rooms*) were perceptually visible. In a similar fashion, we extracted the top 50 ROIs by sorting the z-score of accuracies to obtain the network involved in classifying room memories. Importantly, the room classifiers were trained on the pre-learning room template patterns for the (N-1) group to predict the recalled room during the held-out participant's post-learning object videos. This ensured that 1) the held-out participant's own room templates were never used for testing, avoiding circularity and 2) the room templates of the group were sourced before any room-object associations were learned, eliminating the potential for these room templates to be contaminated by object information.

Revised Results 1 (starting on line 149):

...We identified a set of regions whose pre-learning room reliability was predictive of object reinstatement during verbal recall, including precuneus, posterior parietal cortex, and prefrontal cortex— specifically, superior frontal gyrus. Importantly, using a model comparison analysis, we also found that some of these regions provided a participant-specific predictive benefit, including posterior parietal cortex, posterior ventral temporal cortex, and superior frontal gyrus (see Fig 6B). Lastly, to identify whether room reliability supported object reinstatement *indirectly* by promoting room reinstatement at recall, we conducted a partial correlation analysis controlling for room reinstatement. Even after statistically controlling for room reinstatement, the relationship between room reliability and ROCN object reinstatement remained significant (see Fig 6C). Furthermore, no areas showed a significant decrease in the size of this relationship when we controlled for room reinstatement (see **Partial correlation analysis controlling for room reinstatement** Methods section).

Revised Results 2 (new paragraph starting on line 254)

Lastly, to determine whether room reliability's relationship with object reinstatement was driven by room reinstatement, we ran a partial correlation analysis where we regressed room

reinstatement scores in RRCN from both ROCN object reinstatement and pre-learning room reliability, and then correlated the residuals. After controlling for room reinstatement at retrieval, the relationship between room reliability and ROCN object reinstatement evidence remained significant (Fig 6C). The pattern of results across the brain shown in Fig 6C (when we controlled for room reinstatement) was almost identical to the pattern of results shown in Fig 6A (when we did not control for room reinstatement), and there were not any areas where the effect significantly differed between the two maps. Taken together, these results indicate that fluctuations in room reinstatement during retrieval were not responsible for the effects shown in Fig 6A. For completeness, we also did this for POCN object reinstatement; similarly to what we found for the ROCN, after controlling for room reinstatement at recall, the relationship between room reliability and POCN object reinstatement remained significant, and there were not any areas where this relationship significantly decreased when we controlled for room reinstatement (see Fig 6-Supp1C).

Revised Discussion 1 (starting on line 315):

...For the first time, we were able to show that pre-learning room reliability – the representational quality of an “empty” memory scaffold – is predictive of post-learning object reinstatement in two types of verbal recall; we further demonstrated that – in some regions – a participant’s idiosyncratic room reliability values provided a predictive advantage, beyond what could be predicted by knowing (in general) which rooms were most reliably represented across participants. Finally, we showed that this relationship between room reliability and object reinstatement persists even after statistically controlling for room reinstatement at recall. By ruling out the alternative hypothesis that fluctuations in room reinstatement are (fully) driving the effect, this control analysis provides indirect evidence in support of our preferred hypothesis – namely, that reliable room representations scaffold memory for objects by facilitating the binding of objects to rooms at encoding.

Revised Discussion 2 (new paragraph starting on line 390):

What underlying mechanisms explain the relationship between object reinstatement and room reliability? Our hypothesis was that reliable room representations scaffold memory for objects by facilitating the binding of objects to rooms at encoding. Our finding that room reliability (measured prior to encoding) correlates with object reinstatement (measured during recall) is compatible with this “facilitated binding at encoding” hypothesis. An alternative possibility is that successful reinstatement of reliable rooms during recall promotes object reinstatement for these rooms; this could give rise to a correlation between room reliability and object reinstatement, even in the absence of facilitated binding at encoding. We addressed this alternative hypothesis by controlling for room reinstatement during verbal recall, and found that the relationship between room reliability and object reinstatement remained significant; furthermore, there were not any areas that showed a significant decrease in the size of the effect when we controlled for room reinstatement. The results of this control analysis provide indirect support for our hypothesis that room reliability supports improved room-object binding at encoding; namely, a reliable spatial context representation may provide a stable schematic map that facilitates the integration of new episodic content – the more reliable the container, the easier it is to populate it with information. Future work in which participants are scanned during object-location

encoding would help shed additional light on how room reliability enhances the creation of episodic memories.

New Methods for Partial Correlation (starting on line 989)

Partial correlation analysis controlling for room reinstatement:

To test whether room reliability predicted subsequent object reinstatement when controlling for room reinstatement at recall, we conducted a partial correlation analysis. Specifically, we asked whether the correlation between room reliability and object reinstatement (Figure 6A) remained significant after regressing room reinstatement at recall out of both of these other variables.

To do this, we first constructed a Retrieved Room Classifier Network (RRCN). As described in the **Network Selection Procedure** Methods section, we followed a similar approach to identify ROCN and POCN. In brief, we used a leave-one-participant-out cross-validation procedure in which we classified room recall during the held-out participant's perception of object videos. The top 50 best performing searchlights were used to define the RRCN, which was then used as a mask to extract room reinstatement evidence for our partial correlation analysis.

In this analysis, we wanted to control for room reinstatement that occurred on timepoints when participants verbally recounted room details *and* on timepoints when they verbally described the objects that were paired to a particular room – in principle, room reinstatement during either set of timepoints could be acting to scaffold object retrieval. To this end, we computed two separate room reinstatement scores within the RRCN:

RRCN-room-recall: Room evidence extracted with the RRCN mask during timepoints in which participants were speaking about a room during free and guided recall.

RRCN-object-recall: Room evidence extracted within the RRCN mask during timepoints in which participants were speaking about the object that had been associated with a given room during guided and free recall.

To isolate the unique relationship between room reliability and object reinstatement, we regressed out both RRCN measures from each variable. Specifically, we fit a linear model with ROCN object reinstatement as the dependent variable and both RRCN-room-recall and RRCN-object-recall as predictors. The ROCN residuals from this model represented object reinstatement variance unexplained by room reinstatement. Similarly, we fit a second linear model with room reliability as the dependent variable and the same two RRCN measures as predictors. The room reliability residuals from this model represented room reliability variance unexplained by room reinstatement. Finally, we computed a Pearson correlation between these two residuals. To test for significance, we ran a nonparametric permutation test in which we shuffled the ROCN residuals and recomputed the correlation 1000 times to generate a null distribution of correlation values before running FDR correction for $q < 0.05$.

To identify regions where the relationship between room reliability and object reinstatement had a significant positive or negative change *after* controlling for room reinstatement, we ran a contrast in which the correlation values of the partial correlation were subtracted from the correlation values of our original model. To test for this difference, we computed a composite score of each correlation by averaging the results of each searchlight across recall task types

(i.e., guided and free recalls) and participants. Next, we computed the difference between the results of our original model and the partial correlation as well as on their permutations to get a null distribution of differences. To test for significance, we ran a nonparametric permutation test where we compared the true differences from the null-distribution of differences and FDR corrected for $q < 0.05$.

New Figure 4 - Supplement 2:

Figure 4 - Supplement 2: Retrieved room classifier network (RRCN) methodology and surface maps. (A.) The characteristic room patterns of the $N-1$ group – evoked during a separate phase of the study in which participants viewed room videos before learning room-object associations – were used to train a multinomial logistic classifier. This classifier was then applied to each timepoint on the left-out participant's object-video room recall data. In the pictured example, the left-out participant, Fernando, is recalling the hexagon room that was paired with the carrot object currently being presented. The room classifier, trained on patterns evoked when other participants viewed the rooms, was applied to each timepoint of Fernando's

carrot viewing. We then measured the fraction of timepoints during the object video that were classified as activating the hexagon representation. (B.) After room classification was performed for both post-learning object video runs for each participant, average classification accuracies across participants for each searchlight were then averaged across both runs and then z-scored relative to a null distribution. The top 50 searchlights were then selected to form the RRCN. (C.) Average object classification accuracy during object encoding (thresholded to show searchlights with above-chance accuracy). (D.) Retrieved Room Classifier Network. The surface map shows the top 50 best room classifying searchlights during both post-learning object video tasks across participants.

Comment 3: Additional reporting and clarification on behavioral data

Comment:

“Secondly, the authors could enhance the study by providing a more detailed analysis of behavioral data to reflect the quality of object memory. Unfortunately, the behavioral data analysis appears brief, with only one sentence mentioning behavioral results: “Across both recall types, participants’ recollections were at ceiling, with 92% and 76% of participants achieving over 90% recall accuracy for guided and free recalls, respectively.” Clarification on what constituted a correct response would be helpful since participants were asked to describe room and object details, and yet, these recalled details were not reported. Additionally, 76% of participants scoring over 90% might not be categorized as a ceiling performance.”

Response:

Thank you for this thoughtful comment – we agree that more detail on the behavioral data will strengthen our paper. To this end we have added two new figures (see Figure 3 and Figure 3 - Supplement 1) to expand on our behavioral results.

Figure 3, and its caption, now describes in detail how we computed behavioral recall accuracy for Guided and Free Recalls. Given this comment and another by Reviewer #3, we have also modified the Methods section that describes how object reinstatement evidence was calculated, to clarify that object evidence was computed regardless of whether an object was behaviorally recalled with its corresponding paired room.

Regarding the term “ceiling”, we appreciate the reviewer’s point and now present bar plots in Figure 3 illustrating the impressively high accuracy across participants on both guided and free recall tasks. We hope the visualized accuracy distributions and low variance support our claim that participants generally performed at or near the ceiling on both recall tasks.

For completeness, Figure 3 - Supplement 1 includes additional behavioral data that might be of interest to the reader. The results presented therein have also been summarized in the revised Results section on behavior and a new paragraph was added to the “Post-learning free recall and guided recall” Methods section.

Note: In the original manuscript, we wrote that 92% and 76% of participants scored higher than 90% on guided and free recalls, respectively. The accuracy values for free recalls, unlike those for guided recalls, were based on combined scoring of rooms and objects. In response to your feedback, we revised our scoring approach to focus exclusively on object recall accuracy to better reflect the object-specific ceiling effects while also making the behavioral scoring comparable between guided and free recalls. Under this revised scoring, 92% and 80% of participants scored higher than 90% on guided and free recalls, respectively.

Revised Methods 1 (starting on line 911):

...For any specific guided or free recall run, we collected the total object evidence across all timepoints a participant verbally recalled that object, regardless of whether the associated room was also recalled...

Revised Methods 2 (new paragraph starting on line 745):

Time spent recalling rooms or objects: To assess whether certain rooms or objects were discussed significantly more than others during recall, we conducted an across-participant global mean comparison. For each room and object, we computed the mean time spent speaking across participants. We then performed a one-sample t-test for each room (or object), testing whether its average recall time significantly deviated from the grand mean (i.e., the average across all rooms or objects). Next, we applied a Bonferroni correction to the resulting p-values to account for multiple comparisons.

New Figure 3:

Figure 3: Behavioral recall scoring. On the second day after learning room-object associations in VR, participants went back into the scanner where they performed a free recall and 11 guided recall tasks. In the free recall task, participants were asked to recall and describe with as much detail as possible the rooms and the objects paired to them. In contrast, during the guided recall tasks, participants were presented with 5 contiguously connected rooms and asked to describe the rooms and the objects in them. **A. Example guided recall transcription.** Participant recalls were transcribed manually for the onset and offset timestamps of when rooms and objects were recalled. **B. Example transcribed recall event matrix.** Timestamps of the onsets and offsets of participant recalls were then interpolated from seconds into TRs and organized as event matrices that could then be used to index BOLD recall timeseries. Green and yellow bars indicate room and object recalls, respectively. Object recall timepoints were used to calculate object evidence scores in neural analyses (e.g., the timepoints where participants talked about the Darts were used to measure neural object evidence for recall of the Darts) . **C. Guided recall task, pairings, objects recalled and accuracy calculation.** First column: This participant was presented with a 5-room path and asked to sequentially describe the rooms and the objects in them. Second column: Calculation of behavioral accuracy for guided recalls. Participants were scored based on whether they recalled the objects that were paired to the rooms in the presented path. Although this participant recalled 4 objects, only 3 were associated with the corresponding cued 5-room path. For both guided and free recalls, points were awarded based on whether participants recalled the relevant objects at any point in time during the recall period, regardless of the order in which the objects were recalled or whether they were recalled in association with the correct room. **D. Guided and free recall accuracy distributions.** With these scoring schemes, participants were able to recall objects with high accuracy. Participants' recalls were at ceiling, with 92% and 80% of participants scoring higher than 90% recall accuracy for guided and free recalls, respectively.

New Figure 3 - Supplement 1:

Figure 3 - Supplement 1: Additional behavioral measures. (A–D.) Average time participants spent during recall tasks: (A.) Recalling rooms during guided recalls. B. Recalling rooms during free recalls. (C.) Recalling objects during guided recalls. (D.) Recalling objects during free recalls. For A–D, bar charts describe mean recall time across participants, with error bars indicating standard deviation, and dashed gray lines indicating the global mean. Stars above bar plots indicate a significant difference in time from the global mean, for Bonferroni corrected $p < 0.05$. (E.) Contiguous vs noncontiguous mental traversal. When recalling rooms explored in VR, participants either recalled spatially adjacent (contiguous) rooms or jumped between rooms separated by more than one degree (noncontiguous). (F.) Contiguity in free recall. We measured the proportion of contiguous room transitions during free recall. Across participants, spatially adjacent rooms were recalled more often than expected by chance ($t(24) = 14.19, p < 0.001$) suggesting an unprompted bias towards contiguous mental traversal.

Revised Results (starting on line 183):

2.3 Behavioral recall: On the second day, participants performed two types of recall tasks. During the guided recalls (GRs; 11 runs), participants were presented with the names of 5 rooms that followed a path within the virtual palace and were asked to freely recall details of the rooms and the randomly added objects. During the free recalls (FRs), participants were presented with a blank screen and were simply asked to freely recall, in as much detail as possible, the rooms and the added objects. For the guided recalls, we computed accuracy by counting whether a participant recalled the randomly placed objects in that path regardless of whether they were correctly recalled in order of the path or with the correct room-object pairing. In other words, an object was marked as correctly recalled (out of 5) if it was recalled at any point during the trial. Similarly, for the free recalls, regardless of when an object was recalled, we marked an object as correctly recalled (out of 23) if it was recalled at any point during the free recall. Across both recall types, participants' recalls were at ceiling, with 92% and 80% of participants scoring higher than 90% recall accuracy for guided and free recalls, respectively (Figure 3D). We also found that, in both guided and free recalls, participants spent less time speaking about the Empty Room than the across-participant average (Figure 3 - Supplement 1) – likely because the room was empty (other than the randomly placed object) and there was less to recall. We also measured the proportion of contiguous room transitions during free recall. Across participants, spatially adjacent rooms were recalled more often than expected by chance ($t(24) = 14.19, p < 0.001$), suggesting an unprompted bias towards contiguous mental traversal (Figure 3 - Supplement 1F).

New Methods (new paragraph starting on line 753):

Contiguity in free recall: To assess whether participants tended to recall spatially connected rooms in sequence, we computed, for each participant, the proportion of times each room transition was to an adjacent room (i.e., graph distance of 1 in the adjacency matrix of the virtual environment). Self-loops, where the same room was recalled consecutively, were excluded. To calculate the baseline probability that a participant may have recalled an adjacent room just by chance, for each transition, we counted the number of currently adjacent rooms divided by the 22 possible other rooms (excluding the current room) and then averaged across all transitions. To test for significance, we ran a paired sample t-test where we compared each participant's proportion of contiguous recall with their chance baseline (see Figure 3 - Supplement 1F).

Comment 4: Additional figures and presentation of results

Comment:

“Thirdly, further elaboration on the presentation of results would be beneficial. Beyond the numbers provided for behavioral performance, the results section lacks comprehensive data presentation. It would be helpful to include statistics, bar graphs, scatter plots, and individual participant data. Information on room representation reliability and reinstatement scores, as well as the strength of their correlation, would greatly aid in evaluating data quality.”

Response:

We agree that, in addition to including more statistics describing our behavioral results, further presentation of our neural results would also strengthen our paper. Although infeasible to present individual subject data for all 1483 searchlights (per hemisphere), we have, however, added an entirely new figure (see Figure 6 - Supplement 3) that provides scatterplots illustrating the relationship and its strength between room reliability and object reinstatement for all 25 subjects for an example searchlight.

We have made a small revision in the Results referencing this plot for the interested reader.

New Figure 6 - Supplement 3:

Figure 6 - Supplement 3. Correlation plots between room reliability and object reinstatement in ROCN for an example searchlight. A. Scatterplot illustrating the correlation for an example searchlight and participant. For each participant's free recall (FR) data, we computed the correlation between room reliability and ROCN object reinstatement evidence in the ROCN mask. **B.** Individual scatterplots for all 25 participants showing the relationship between room reliability scores and ROCN reinstatement evidence, extracted from the example searchlight seen in A. Plots are sorted by participants, moving from left to right in order of highest to lowest correlations, respectively. Dashed lines in scatterplots represent the identity

line ($y = x$), and Pearson's r correlation values are reported on the title-line of each plot. Room reliability and ROCN object reinstatement scores were z-scored to facilitate visualization of relationships.

Revised Results (starting on line 241):

...Specifically, at each searchlight, we correlated each participant's room reliability with their own composite ROCN object reinstatement score (Fig 6A; see Fig 6 - Supp 3 for an example searchlight)...

Comment 5: Clarification on participant-specific benefit analysis

Comment:

"Fourthly, in the context of participant-specific versus group-averaged results, regions identified with superior prediction using the original model were discussed. However, it remains unclear whether "better" corresponds to statistically or numerically greater predictions, and clarification on these differences would be valuable."

Response:

Thank you for pointing this out. We appreciate the opportunity to clarify the interpretation of "better" in the context of our model comparison. In the revised manuscript, we clarify that "better" refers to a statistically significant improvement in model performance, as determined by nonparametric permutation testing on the difference in R^2 values between models.

Moreover, based on a thoughtful suggestion from Reviewer #3, we have replaced our original model comparison with a more sensitive approach where we compare model performance (as measured by R^2) of our original model (i.e., the participant-specific model) with every other participant's model individually.

We have updated our Methods and Results with this new analysis and, as noted above, we now clearly state that these model comparisons are statistically better (not just numerically better), as measured by our nonparametric permutation tests on the differences between R^2 s. The relevant figure has been updated accordingly (now in Figure 6).

To see these changes please refer to our response to Reviewer #3 - Comment 2.

Comment 6: Lack of evidence for scaffolding and relationship to the method of loci (MOL)

Comment:

"Lastly, the study offers limited mechanistic explanations for the observed correlations. For instance, evidence supporting scaffolding is lacking. The method of loci (MOL) emphasized in this study involves learning the sequence of rooms. It would be insightful to explore how the learning of sequence order

influences the reliability and distinctiveness of rooms and object recall. Furthermore, while examining whether this correlation applies to any paired association task or is specific to the method of loci (MOL) might exceed the current scope, discussing this aspect could offer valuable insights.”

Response:

Thank you for bringing this concern to our attention. We agree that it's crucial that we clarify the mechanistic role of spatial contexts and how, in our current work, they scaffold memory. In response to your comment (which echoed concerns from other reviewers), we have clarified our primary hypothesis about mechanism – namely, that reliable room representations scaffold memory for objects by facilitating the binding of objects to rooms at encoding. Our finding that room reliability (measured prior to encoding) correlates with object reinstatement (measured during recall) is compatible with this “facilitated binding at encoding” hypothesis. In the revised manuscript, we also discuss an alternative explanation for this result – namely, that room reliability (measured prior to encoding) could promote successful room reinstatement during recall periods, which then promotes object reinstatement; this could give rise to a correlation between room reliability and object reinstatement, even in the absence of facilitated binding at encoding. In the revised Introduction, we now introduce both of these possibilities, to set the stage for our updated results.

Revised Introduction (new paragraph starting on line 99):

..this implies that having a stable and distinctive neural representation for a location *before* associating an object to that location will be predictive of subsequent reinstatement for that object representation.

Our primary mechanistic hypothesis for *why* this would occur was that reliable room representations facilitate the binding of room to object information at encoding (e.g., the sturdier a wall is, the easier it is to hang a painting on it). However, facilitated binding at encoding is not the only way that having a stable and distinctive room representation could facilitate subsequent object reinstatement; an alternative possibility is that having a stable and distinctive room representation has no effect on room-object binding at encoding, and that instead it boosts object recall *indirectly* by boosting the degree to which the room representation is reinstated at test, which – in turn – boosts reinstatement of associated object information (e.g., the brighter the light in a dark room, the easier it is to see what's inside). We will present the results of analyses that control for this alternative possibility.

To test whether reliable spatial contexts scaffold subsequent memory, we custom-built a virtual...

Additionally, we have strengthened our claims with our partial correlation analysis (Figure 6C). This analysis statistically controls for room reinstatement at recall. If room reliability were simply a proxy for stronger room activation at retrieval (which then cues object recall), controlling for that room reinstatement should eliminate the observed effect. Instead, we found that room reliability continued to predict object reinstatement even after controlling for room activity at retrieval, providing new, although indirect, evidence suggesting that spatial context reliability supports memory at encoding. To see these changes, please refer back to our previous response (Reviewer #1 - Comment 2).

We also appreciate the opportunity to clarify our connection to MoL. While our study is heavily inspired by the method of loci, our work herein diverges from the traditional technique in several ways. First, unlike classical implementations of MoL, our virtual memory palace was not linear or truly sequential, with some rooms linked to more than two others. As a result, our virtual memory palace was more like an undirected graph than a linear one typically associated with MoL. Second, during the learning phase participants were not instructed to encode objects in an explicit sequence of rooms. Instead, they freely explored the virtual environment and encoded in whatever order they chose with the freedom to revisit rooms as many times as needed. Third, participants were not taught nor encouraged to use any particular mnemonic strategy. Rather, they memorized the room-object bindings in whatever way felt natural to them.

Our departure from traditional MOL allowed us to examine how pre-learning spatial context reliability influences memory in a more ecologically valid setting—one that reflects how people naturally revisit and interact with real-world environments (and an environment that was explored by all participants vs the idiosyncratic ones associated with MoL). As a result, our findings offer conclusions that are likely more generalizable to everyday memory formation than those drawn from studies using traditional method of loci implementations. At the same time, we also believe that our results here contribute to the method of loci literature – after all, the theoretical basis for the effectiveness of this technique in binding arbitrary information comes from the idea that well learned spatial locations can be powerful scaffolds for the encoding and retrieval of new information. We have revised the Introduction and Discussion section where we connect our study with MoL to clarify these differences.

Given your astute observation on the sequence-learning nature of MoL, we do think you might be interested in one of the newly added behavioral supplements (Figure 3 - Supplement 1F) where we looked at whether participants recalled rooms in a spatially-connected order during the free recall task. Interestingly, participants tended to mentally travel to adjacent rooms as they recalled the rooms and objects during the free recall task. This suggests that there might be a bias towards sequential recall – even though participants were not explicitly instructed to do so. Because this is not in the scope of our original study, we do not discuss it in the revised manuscript, but leave a reference to it in the Results for the reader to peruse. To see the Methods for this new analysis, please refer back to our response to Reviewer #1 - Comment 3.

Revised Discussion on MOL (starting on line 457)

3.3 Our experimental paradigm and the method of loci: Our “memory palace” paradigm draws inspiration from the mnemonic technique called the method of loci (MOL), in which items are associated with an imagined sequence of spatial locations in a pre-learned map. However, our study diverges from this technique in several key ways: Unlike many implementations of MOL, participants were not required to encode or recall to-be-remembered items in an explicit linear sequence of rooms, nor were they instructed to use any particular mnemonic during room-object binding. Instead, participants explored the virtual environment freely and developed their own strategies for memorization.

Despite these differences, the motivation for this technique is related to the hypothesis tested in this study: that a well-learned spatial map consisting of many distinct locations is the optimal encoding environment for new item memories. The learnability of this technique suggests that it may rely on inherent spatial memory structures shared across people. In fact, the ability to

improve memory through this spatially-based technique has been shown across multiple studies behaviorally and neurally (behavioral: Reggente et al., 2020; Legge et al., 2012; Roediger, 1980; Bower and Reitman, 1972; Mo`e and De Beni, 2005, among many; neural: Dresler et al., 2017; Wagner et al., 2021; Nyberg et al., 2003; Maguire et al., 2003; Mallow et al., 2015; Kondo et al., 2005; Fellner et al., 2016; Liu et al., 2022). Generally, neuroimaging studies of this technique have largely focused on the impact of MOL (at varying levels of training or compared to other mnemonics) during item encoding (Maguire et al., 2003; Nyberg et al., 2003; Dresler et al., 2017; Wagner et al., 2021), with only a few performing univariate contrasts during recall (Kondo et al., 2005; Mallow et al., 2015; Fellner et al., 2016; Liu et al., 2022), and only one, to our knowledge, examining multivariate pattern activity for loci, items, and their conjunctive associations (Huang et al., 2025). The univariate results during recall have shown enhanced engagement of regions including RSC and precuneus after instruction in MOL (Kondo et al., 2005), suggesting that spatial representations of loci are strategically activated during retrieval. A recent study measuring multivariate activity patterns during MOL (Huang et al., 2025) found robust representations for individual loci during the creation and retrieval of item-locus pairs in regions including precuneus and posterior parietal cortex, suggesting potentially overlapping mechanisms in how our naive subjects and MOL-trained individuals use spatial information for item memorization. It remains an open question whether enhanced room reliability helps support memorization when using MOL.

Response to Reviewer #2

We would like to thank the reviewers for their insightful and constructive comments which have greatly improved the manuscript and appreciate the recognition of our design and execution. Please find below your comments and how we addressed them.

Comment 1: Additional literature on the timing of spatial context and item reinstatement

Comment:

“The authors could cover a little more literature on the timing of spatial context reinstatement vs. object reinstatement, this could help bolster the argument in the introduction that spatial context is a scaffold/special”

Response:

Thank you for this suggestion! We agree that covering more literature on the temporal dynamics of spatial context reinstatement during retrieval helps illustrate one of the ways spatial contexts can benefit memory. In fact, your suggestion (as well as Reviewer #1’s) made us realize that we needed to refine our framing of “scaffold” as a process that can support memory at encoding or retrieval (or both). To this end we have revised the Introduction to include additional literature on the temporal dynamics of spatial context and object reinstatement at retrieval, while also clarifying how spatial contexts can play a supportive role during encoding as well.

Specifically, we now cite work showing that spatial contexts can be reinstated prior to or concurrently with item retrieval, providing evidence of their role as effective memory cues during retrieval (Herweg et al., 2020; Miller et al., 2013). Similarly, we now cite work demonstrating that explicit binding and incidental binding of a spatial context with an to-be-remembered item during encoding boosts memory for that item later (Reggente et al., 2020; Shin, Masis-Obando, et al., 2019), providing evidence of how spatial contexts can shape memory during encoding.

Lastly, we have clarified our thoughts on possible mechanisms that could give rise to a relationship between pre-learning room reliability and subsequent object reinstatement. We now explicitly state that our primary hypothesis is that room reliability facilitates room-object binding at encoding, and we contrast that with an alternative hypothesis that pre-learning room reliability supports improved room reinstatement at retrieval (but has no effect on room-object binding at encoding). We have also bolstered the rigor of our paper with a new partial correlation analysis (Figure 6C) that addresses this alternative hypothesis, showing that room reliability’s relationship with object classifier evidence persists when we control for room activation at retrieval.

Overall, we have modified the Introduction (to provide more background and clarification on what we mean by scaffold – including citations to papers on the timing of spatial context cueing effects), the Results (to include the new partial correlation analysis), the Methods (with a description of the new analysis), and the Discussion (to elaborate on these new findings). To see the specific revisions for the

Results and Discussion that we made as a result of our new partial correlation analysis – please refer to our response to Reviewer #1 - Comment 2.

Revised Introduction 1 (starting on line 69):

...This behavioral work is complemented by neuroimaging studies of autobiographical memory showing that spatial contexts have a strong influence on the neural representations of remembered or imagined autobiographical events (Robin et al., 2018; Hebscher et al., 2018; Reagh and Ranganath, 2023, among others; for a review see Hassabis and Maguire, 2007). The networks associated with spatial contexts are maintained during multiple phases of memory retrieval, possibly acting as a scaffold for accessing additional event details (Gurguryan and Sheldon, 2019). For example, spatial contexts can be reinstated prior to or concurrently with the retrieval of an item or episode (Herweg et al., 2020; Miller et al., 2013).

Beyond retrieval, prior theoretical work on episodic memory suggests that – at encoding – features of an ongoing experience are bound to the context in which they occur (Yonelinas, et al., 2020; Ranganath, 2010; McClelland et al., 1995), allowing spatial contexts to serve as structured “containers” that organize and support the integration of new experiences (Gilboa & Marlatte, 2017; Preston & Eichenbaum, 2013). Consistent with this view, explicitly binding objects to their spatial context during encoding enhances subsequent memory for those objects (Reggente et al., 2020). In another recent study (Shin, Masís-Obando et al., 2021), participants encountered words within two distinct spatial contexts (each associated with a separate schema) and judged each word’s relevance to its context without knowing there would be a later memory test. If reinstating the context at retrieval were sufficient to boost memory, all words should have benefitted equally. Instead, only context-relevant words showed a memory advantage, suggesting that these items were more effectively bound to the spatial context during encoding.

Comment 2: Clarification on definition of spatial context

Comment:

“It might help to define spatial context, both broadly when discussing the background literature, and how it is defined experimentally here. If we understood correctly, in this paradigm it is a room with objects and not an empty room (again something we didn’t realize until we got to the methods) and there is music/sounds distinct to that context. This spatial context is then populated with and more prominent (random?) objects for the memory encoding. This feels like a more rich/naturalistic definition of context, but is it similar to what others have done in the past? Does that change anything?”

Response:

Thank you for highlighting the need to clarify how we define spatial context, both conceptually and in the context of our experimental design. As you pointed out, our virtual rooms were not empty (except the “Empty Room” which was intentionally empty). In fact, they were perceptually rich with ambient sounds, soundtracks, colorful interiors, themed and filled with background theme-congruent objects. We

designed them to be as perceptually distinct as possible to reflect the richness of real-world spatial locations and promote strong context-item associations.

We have now clarified in the Introduction, Results and Discussion that we define spatial contexts as the locations in which experiences occur –operationalized in our experiment as the multisensory virtual rooms participants explored.

Importantly, while earlier work on context-item binding used still images of “empty” contexts (i.e., environments without additional congruent objects), there is a plethora of recent work using visually rich non-empty contexts in the schema literature (images: Audrain & McAndrews, 2022; videos: Reagh & Ranganath, 2023), context-dependent memory literature (e.g., Walti et al., 2019; Shin, Masis-Obando et al., 2021), and method of loci literature (e.g., Essoe et al., 2022). Thus, while our definition of spatial context is more naturalistic than earlier psychological work, it is consistent with current approaches aiming to understand memory under more ecologically valid conditions.

Revised Introduction 1 (starting on line 111):

...we custom-built a virtual reality “memory palace” environment of 23 perceptually distinct rooms, each with distinct soundtracks, interiors and room-congruent objects, that participants explored using a head-mounted virtual reality display Fig1.

Revised Introduction 2 (starting on line 40) :

... In what ways can a spatial context (i.e., the location in which an experience takes place) serve as a scaffold for storing and accessing the details of past episodes?...

Revised Results (starting on line 167):

We found significant room reliability across most of the cortex. Unsurprisingly, given the audio-visual nature of the room videos, we found high reliability scores in auditory and visual cortex, as well as in precuneus, and posterior hippocampus (Figure 2C).

Revised Discussion (starting on line 413):

We described the representational stability and distinctiveness of a spatial context through a reliability score that measured the specificity of a room's representation across runs. *These spatial contexts were designed to be visually and auditorily rich to reflect real-world contexts.* Given that room reliability was derived from audio-visual stimuli, it was not surprising to find the strongest reliability in visual and auditory cortex...

Comment 3: Clarification on rationale behind recall tasks

Comment:

“It wasn’t clear to us why they did two different types of recall? (though a strength to find the same results across both)”

Response:

Thank you for bringing this important point of clarification to our attention. Our rationale for using both guided and free recall was to assess the role of spatial context reliability under varying conditions that reflect a range of real-world task demands. Specifically, we included both guided and free recall since realistic verbal recall can sometimes be prompted with cues (as in our guided recall task) and sometimes it can be fully self-initiated (as in our free recall task).

As you pointed out, we found that room reliability predicted object reinstatement in both guided and free recall tasks – providing an internal replication and highlighting the generalizability of our measure across recall types. Given the similarity of these results, and the confusion surrounding the presentation of both results (a point raised by Reviewer #1 as well), we have decided to combine these results into a single figure. Specifically, we averaged the results of Figure 5A/C with Figure 5B/D into a new set of brainmaps (Figure 6A and Figure 6B). We tested for significance by running permutation tests on their combined null and ran FDR-correction on the resulting p-values. We have modified the Results and Methods to account for this change and summarize it briefly in the Discussion when interpreting the robustness of the effects across task types (see revision below). To see the other changes please refer back to Reviewer #1 - Comment 1 and Comment 2.

Revised Discussion (starting on line 346):

In general, we found that object reinstatement was predicted by room reliability in precuneus, RSC, insula, frontal cortex, and regions throughout lateral parietal cortex (Fig 6) suggesting that measuring the structural integrity of a spatial context representation before a life-episode is predictive of how well that episode will be reinstated later. Moreover, these effects were found separately for both guided and free recall, providing an internal replication of our results and suggesting that stable context representations are useful for retrieval across multiple kinds of memory tasks.

Comment 4: Relationship between room features and room reliability

Comment

“Anything special in terms of features about the rooms that were most reliable across participants? How much individual variability?”

Response

Thank you for highlighting a fascinating analysis our paradigm lends itself to. In addition to you, Reviewer #3, and we, too, were curious about which room properties related to our room reliability measure, and we were eager to run these new analyses. Are bigger rooms more reliable? Do more reliable rooms tend to have more objects in them? Does the connectedness of a room (how many rooms it's connected to) relate to reliability? We measured 6 different features: “degree” (how many rooms are connected to room of interest), “ratio of volume occupied” (the proportion of volume occupied by objects inside a room), “number of objects” (manual count of every object inside a room), “area” (area covered by room floor), “number of corners” (sum of wall corners in room), and “has window”

(binary of whether this room has a view to the outside). In brief, in default mode network regions, the most reliable rooms tended to be those that were small in size, had many corners, and had a window with a view to the outside.

We have included a new Methods section, a Results section, and a supplementary figure (Figure 2 - Supplement 1) illustrating the associations between these room features and room reliability.

New Methods Section (starting on line 1037):

Relationship between room reliability and room features: Do properties of a room contribute to the reliability of their representation? We sought to identify whether physical or graph theoretical features of a room contributed to the reliability of their representation. To do this we used the 3D Unity model of the environment to compute a list of physical features such as total room volume, total volume occupied by background objects, the proportion of occupied volume and total room volume, area, object count, number of corners, and whether the room has a window (i.e., a view to the outside), and used the room adjacency matrix to compute graph-theoretical features such as degree, betweenness, closeness, eigenvector, and pagerank. We then selected six features (degree, ratio of occupied volume, background object count, floor area, number of corners, and “has window”) that were the least collinear and provided conceptually non-overlapping properties (e.g., betweenness and degree are collinear). We z-scored each feature (except the binary “has window”) and then ran a searchlight analysis where we regressed room reliability on each of the six z-scored features for every participant. To test for statistical significance of each of the resulting beta coefficients, we ran a nonparametric permutation test where room reliability was shuffled 1000 times within participants before regressing again on the features to generate a null distribution of beta coefficients. We then averaged across participants before running FDR-correction on the resulting z-values and thresholding at $q < 0.001$.

New Results (new paragraph starting on line 170):

Are there particular room properties, such as size, complexity, or connectedness that contribute to the reliability of room representations? To identify which room features contribute to room reliability, we ran a searchlight analysis where, within each searchlight, we ran a multiple regression predicting room reliability based on six different room features; we generally found that, in default mode network regions, the most reliable rooms tended to be those that were small, had many corners, and had an opening with a view to the outside (Figure 2 - Supplement 1).

New Figure 2 - Supplement 1:

Figure 2- Supplement 1: Relationship between room reliability and room features. (A.) Regression schematic predicting room reliability with room features. Six different room features were chosen to predict room reliability. From left to right: “degree” (how many rooms are connected to room of interest), “ratio of volume occupied” (the proportion of volume occupied by objects inside a room), “number of objects” (manual count of every object inside a room), “area” (area covered by room floor), “number of corners” (sum of wall corners in room), and “has window” (binary, indicating whether this room has a view to the outside). (B.) Significant room feature regression coefficients. In a searchlight analysis, reliability for a room (in that searchlight) was predicted by six different room features for each participant. Statistical significance for the resulting beta coefficients was determined by a non-parametric permutation test and FDR-corrected for $q < 0.001$. All six surface maps are colored based on the magnitude of significant z-scored coefficients, with blue showing negative and red showing positive relationships, respectively.

Comment 5: Speculation on similarities/differences between emotional states and spatial context states

Comment:

“Could you do the same study/analysis for emotional states? Would you expect the same results?”

Response:

Thank you for this fascinating thought experiment. Our expertise is not in emotion, but we appreciate the opportunity to consider a potential extension to our experiment.

Our current understanding is that spatial contexts serve as discrete, structured “containers” that facilitate efficient associative encoding of new memories. It is plausible that emotional states may similarly act as containers for memories tied to valenced events (e.g., a birthday party, the birth of a baby, traumatic event).

However, there are differences that could complicate an analogous study using an emotional state as the primary context. In this work, we tightly controlled the creation of a virtual environment with distinct and salient perceptual features to form participants' spatial context representations. An equivalent 'emotional palace' presents a significant experimental challenge because it would be difficult to discretize and reliably induce a large number of distinct emotional states analogous to our 23 spatial contexts. Moreover, in our paradigm, during verbal recall, participants were endogenously cueing their memory to reconstruct their recent experiences. Thus, having participants internally cue emotional states for memory retrieval in a comparable way also presents a hurdle. This might be feasible in populations such as trained actors, particularly method actors, whose practice involves deliberate emotional generation and regulation. A study with method actors would require careful design, but nonetheless would be interesting if done effectively.

Finally, prior work suggests that highly arousing or traumatic events may engage distinct encoding mechanisms from those observed in non-traumatic events. For example, a traumatic event may disrupt hippocampal processing during encoding, leading to fragmented and decontextualized memories (e.g., Brewin, Gregory, Lipton, & Burgess, 2010). In turn, these fragmented memories are more prone to intrusion in inappropriate contexts. In this case, emotional states may hinder rather than enhance episodic memories. Thus, running an experimental paradigm with an emotional palace where a participant deliberately recalls prior experiences while tightly controlling the valence of emotional states to avoid confounding mechanisms presents an additional challenge. That is not to say that it would be impossible to run this study – this would, in fact, be very interesting to explore in future work.

Comment 6: Speculation on how novelty and familiarity of a spatial context might influence room reliability

Comment:

“A discussion on the relationship between familiarity/novelty and context, and how that maps on to reliability/distinctiveness/memory benefits could be helpful. Is the novelty effect in memory separate from spatial context? Or do we get a benefit because novel spatial contexts are distinct (even if not reliable) on account of their novelty? Or does it need to be novel and dissimilar in experience (Asian city block vs. different US city block for a person from the US who has never been to Asia). How do repeated spatial contexts stay distinct and useful? Any plans to have more long-term familiarity tests?”

Response:

Thank you for raising this thought-provoking question about the interplay between familiarity, novelty, distinctiveness, and their relationship to memory and spatial context. While our current study was not explicitly designed to disentangle these variables, we agree that these are critical factors that warrant further investigation.

In our work, participants were thoroughly familiarized with each of the rooms before the scanning sessions, and the memory effects we observed are therefore unlikely to be related to novelty. That said, your point is well-taken: novelty and distinctiveness often co-occur, especially early in learning, and may both contribute to memory encoding. A novel context might boost attention or encoding efficiency initially, but its benefit may decline with repetition unless its representation remains distinctive and stable over time. This raises important questions about the temporal dynamics of novelty versus reliability: Do initially novel contexts lose their effectiveness as scaffolds once they become familiar, or do they stabilize into even more reliable contexts that *continue* to support memory? Your point of real-life contexts is compelling – even after returning to the same environments over and over again, we are still able to form distinct episodic memories within them. This may be due to new sub-events, new emotional states (as your prior comment alluded to), weather, etc, that enrich the associations of the new episode to that spatial context. While our study focused on one episode per room (the newly placed object), we agree that future longitudinal work should explore how room reliability supports encoding and retrieval across multiple episodes within the same spatial contexts. We have revised the Discussion section to bring this point to light, and potentially inspire readers for compelling future work.

Revised Discussion (starting on line 447):

However, in our paradigm, participants were explicitly using a context-based strategy for retrieving items, mentally simulating rooms and trajectories through rooms in order to reinstate item memories. In this case, we would expect that having a reliable contextual index for episodic memories would be critical for effective recall of items, consistent with our findings that stability in scene-related brain regions predicted item reinstatement. Future work could investigate whether this relationship disappears or reverses in other situations, such as **when** many items are paired with the same room (reducing the usefulness of rooms as memory cues), **when** rooms **have** features that **vary**, e.g., with time of day (such that representational variability might reflect meaningful changes in contextual features), **or when the recall task requires reporting only objects while** suppressing **recall of room features**. Similarly, novelty may influence how

room reliability scaffolds memory: a new context may be less stable than a highly-familiar location, but could still enhance memory because its novelty promotes additional attention and processing. Future work examining how repeated exposure and contextual novelty interact with reliability could shed new light on their contributions to memory.

Minor Comments: Clarification on auditory content in rooms and word choice

Comment:

“Auditory cortex is one of the most reliable brain areas for room reliability representations. This seems surprising, until you read on in the methods and find out they played distinct music/soundtracks for the room. Might want to pop that in earlier.”

Response:

Thank you for highlighting this potential point of ambiguity. As per your previous comment: we have clarified that our rooms are our spatial contexts and were designed to be as perceptually distinct as possible. We have added this clarification in the Introduction and in the Results section:

Revised Introduction (starting on line 111):

“...we custom-built a virtual reality “memory palace” environment of 23 perceptually distinct rooms, each with distinct soundtracks, interiors and room-congruent objects, that participants explored using a head-mounted virtual reality display Fig1.”

Revised Results (starting on line 167):

We found significant room reliability across most of the cortex. Unsurprisingly, given the audio-visual nature of the room videos, we found high reliability scores in auditory and visual cortex, as well as in precuneus, and posterior hippocampus (Figure 2C).

Comment:

“line 57 “may enhance” does it or not? Is the literature mixed?”

Response:

Thank you for bringing this to our attention. We tend to exercise caution whenever making absolute claims about research findings. However, the referenced papers provide compelling evidence that spatial context cues result in more detailed recall than other cues. The “may” has been removed.

Revised Introduction (starting on line 53):

Recent behavioral work has also suggested a privileged role for spatial contexts as cues for memory retrieval. For example, spatial context cues: a) enhance episodic recall when compared with temporal, thematic (e.g., romantic experience), person, or objects cues for imagined or real autobiographical memories...

Response to Reviewer #3

We thank the reviewer for their insightful comments and enthusiastic and positive evaluation of our work and its VR-fMRI methodology. Your comments have greatly improved our revised manuscript. Please find our responses to your comments below:

Comment 1: Cognitive mechanism behind room reliability

Comment:

“What do you think is the cognitive mechanism behind the link of room reliability and object reinstatement? Is it that more reliable rooms serve as better cues to an association with the objects?”

Response:

Thank you for the opportunity to expand on our results and how they might speak to room reliability's influence on object reinstatement. It is clear from your comment, along with Reviewer #1 and #2's comments, that we needed to clarify this essential interpretive piece to our results. What do we mean by scaffold?

In the revision, we have clarified our thoughts on possible mechanisms that could give rise to a relationship between pre-learning room reliability and subsequent object reinstatement. We now explicitly state that our primary hypothesis is that room reliability facilitates room-object binding at encoding, and we contrast that with an alternative hypothesis that pre-learning room reliability supports improved room reinstatement at retrieval (but has no effect on room-object binding at encoding). We have also included an entirely new analysis to address this potential ambiguity in the interpretation of our results. In this analysis, we statistically controlled for room reinstatement at recall. Using a newly defined room-specific mask (i.e., Retrieved Room Classifier Network; RRCN) from a separate portion of the experiment, we regressed out room reinstatement from ROCN object reinstatement scores and room reliability, and then correlated the residuals. If the observed link between room reliability and object reinstatement were entirely driven by room-based retrieval cues (i.e., if more reliable rooms were simply more likely to be reinstated and thus cue object recall), the effect should go away when we regress out room reinstatement during recall. However, the effect remained robust, thereby providing indirect evidence for our primary hypothesis (facilitated binding at encoding). This control analysis does not rule out the possibility of an influence of room reliability on room reinstatement, but it shows that this can not be the *only* factor driving the observed relationship between pre-learning room reliability and subsequent object reinstatement.

Please find below our revised Introduction, Results, and Discussion in which we now address these two hypotheses.

Revised Introduction (new paragraph starting on line 99): *

..this implies that having a stable and distinctive neural representation for a location *before* associating an object to that location will be predictive of subsequent reinstatement for that object representation.

Our primary mechanistic hypothesis for *why* this would occur was that reliable room representations facilitate the binding of room to object information at encoding (e.g., the sturdier a wall is, the easier it is to hang a painting on it). However, facilitated binding at encoding is not the only way that having a stable and distinctive room representation could facilitate subsequent object reinstatement; an alternative possibility is that having a stable and distinctive room representation has no effect on room-object binding at encoding, and that instead it boosts object recall *indirectly* by boosting the degree to which the room representation is reinstated at test, which – in turn – boosts reinstatement of associated object information (e.g., the brighter the light in a dark room, the easier it is to see what’s inside). We will present the results of analyses that control for this alternative possibility.

To test whether reliable spatial contexts scaffold subsequent memory, we custom-built a virtual...

Revised Results (new paragraph starting on line 254)

Lastly, to determine whether room reliability’s relationship with object reinstatement was driven by room reinstatement, we ran a partial correlation analysis where we regressed room reinstatement scores in RRCN from both ROCN object reinstatement and pre-learning room reliability, and then correlated the residuals. After controlling for room reinstatement at retrieval, the relationship between room reliability and ROCN object reinstatement evidence remained significant (Fig 6C). The pattern of results across the brain shown in Fig 6C (when we controlled for room reinstatement) was almost identical to the pattern of results shown in Fig 6A (when we did not control for room reinstatement), and there were not any areas where the effect significantly differed between the two maps. Taken together, these results indicate that fluctuations in room reinstatement during retrieval were not responsible for the effects shown in Fig 6A. For completeness, we also did this for POCN object reinstatement; similarly to what we found for the ROCN, after controlling for room reinstatement at recall, the relationship between room reliability and POCN object reinstatement remained significant, and there were not any areas where this relationship significantly decreased when we controlled for room reinstatement (see Fig 6-Supp1C).

Revised Discussion 1 (starting on line 315):

...For the first time, we were able to show that pre-learning room reliability – the representational quality of an “empty” memory scaffold – is predictive of post-learning object reinstatement in two types of verbal recall; we further demonstrated that – in some regions – a participant’s idiosyncratic room reliability values provided a predictive advantage, beyond what could be predicted by knowing (in general) which rooms were most reliably represented across participants. Finally, we showed that this relationship between room reliability and object reinstatement persists even after statistically controlling for room reinstatement at recall. By ruling out the alternative hypothesis that fluctuations in room reinstatement are (fully) driving the effect, this control analysis provides indirect evidence in support of our preferred hypothesis – namely, that reliable room representations scaffold memory for objects by facilitating the binding of objects to rooms at encoding.

Revised Discussion 2 (new paragraph starting on line 390):

What underlying mechanisms explain the relationship between object reinstatement and room reliability? Our hypothesis was that reliable room representations scaffold memory for objects by facilitating the binding of objects to rooms at encoding. Our finding that room reliability (measured prior to encoding) correlates with object reinstatement (measured during recall) is compatible with this “facilitated binding at encoding” hypothesis. An alternative possibility is that successful reinstatement of reliable rooms during recall promotes object reinstatement for these rooms; this could give rise to a correlation between room reliability and object reinstatement, even in the absence of facilitated binding at encoding. We addressed this alternative hypothesis by controlling for room reinstatement during verbal recall, and found that the relationship between room reliability and object reinstatement remained significant; furthermore, there were not any areas that showed a significant decrease in the size of the effect when we controlled for room reinstatement. The results of this control analysis provide indirect support for our hypothesis that room reliability supports improved room-object binding at encoding; namely, a reliable spatial context representation may provide a stable schematic map that facilitates the integration of new episodic content – the more reliable the container, the easier it is to populate it with information. Future work in which participants are scanned during object-location encoding would help shed additional light on how room reliability enhances the creation of episodic memories.

Comment 2: Adjustment of participant-specific benefit analysis

Comment:

“In the main text, you discuss participant-specific reinstatement regions, but in the Methods you note the importance of considering the group-wise regions too--they have different implications for how to think about what makes a good memory palace. Currently it's hard to sense the extent of the group-wise effect because right now the group-wise effect is using N-1 participants (vs. 1 participant for the participant-wise effect), so by nature of having more participants it may show stronger effects. I wonder if there's an analysis that could show what the specific group-wise effect would be -- for example by iteratively using one participant's room reliability to predict another participant's object classifier evidence, averaged across many iterations. (So the participant-specific and group-wise measures both use only one participant's data per task at a time).”

Response:

Thank you for this fantastic suggestion which has greatly improved this manuscript. We agree that in creating a model using the average reliability of N-1 participants we may have been over-representing the group. We have implemented your suggestion of a fairer model comparison where every participant's model is compared iteratively to every other N-1 participants' models. The Methods and Results sections (including figures) have been updated with the new and improved analysis results.

Revised Methods (starting on line 967):

To assess whether the observed within-subjects relationship between room reliability and object reinstatement has a participant-specific component, we compared the predictive performance of an ordinary least squares regression derived from a participant's own room reliability values with one based on other individuals' reliability values. Specifically, for the participant-specific model (as in Fig 6A), we calculated the coefficient of determination (R^2) of a model where the participant's object reinstatement probabilities were predicted by their own room reliability values. For the group-wise model, we iteratively predicted that participant's reinstatement probabilities from every single other participant and averaged the resulting N-1 R^2 values. We then ran a model comparison test where we took the difference between the R^2 of the participant-specific model and the average R^2 from the other-participant prediction models. A significant positive difference in this analysis indicates that the participant-specific model explains the variability in object evidence better than other individuals (and thus the observed results can not be entirely due to the group-wise effect). To test for statistical significance, we ran a nonparametric permutation test where the object labels were randomly shuffled 1000 times to generate a null distribution of model performance for each model. To generate a single composite map summarizing model performance across recall tasks, we averaged R^2 values across guided and free recalls for each model separately, computed the difference in R^2 , and tested for significance using a nonparametric permutation test on the combined null distribution of R^2 differences, followed by FDR-correction on the resulting p-values. For completeness, separate results for guided and free recall are provided in Figure 6 – Supplement 2, while the main results reflect the combined composite analysis (see Figure 6).

Revised Results 1 (starting on line 270):

To what extent do the effects in Fig 6A reflect group-level differences across rooms (whereby some rooms have both high reliability and high item reinstatement in all participants) versus *participant-specific* differences in which rooms are most reliable in their individual mental maps? To answer this question, we compared the coefficient of determination (R^2) between 1) our original participant-specific model, where each participant's object classifier evidence was predicted using their own room reliability values, and 2) the average R^2 of N-1 models where – in each model – the left-out participant's object classifier evidence was predicted using a different participant's room reliability values (i.e., one model for each of the N-1 other participants). We then took the regions where there was a positive and statistically significant participant-specific effect (i.e., better prediction with the original model), and intersected them with the correlational analysis performed in Fig 6A.

Revised Results 2 (281):

...This process revealed a participant-specific benefit of room reliability in posterior parietal cortex (near angular gyrus), insula, and superior frontal gyrus (Fig6B). Interestingly, there was also a participant-specific effect where room reliability in a small section of right parahippocampal cortex was negatively associated with ROCN reinstatement evidence....

Revised Results 3 (starting on line 293):

... Across both recall tasks, there was a participant-specific benefit of room reliability in posterior parietal cortex, posterior medial cortex, right insula, and portions of right lateral superior and frontal gyrus (Fig6 - Supp1; refer to Figure 6 - Supp 2 for guided and free recalls separately)...

Comment 3: Relationship between room features and room reliability and expansion of object reinstatement results

Comment:

“Since a part of this study’s central claim is that some rooms create more robust representations than others, I’d be curious to know--what were the rooms that were worse/better? Were there any features that might relate to their reliability, like room size, number of connecting rooms, or number of background objects? (The latter is helpful to test because it is posed in the discussion, too). Were there any objects that also tended to have more evidence over others across rooms? (Though this is not as central a part of the study--am just curious if it goes both ways.”

Response:

Thank you for bringing this to our attention – both you and Reviewer #2 highlighted that identifying what properties of a room contribute to room reliability would strengthen our paper. As mentioned above in our response to Reviewer #2, we too were very curious and excited about this, and we have added a new supplementary figure (Figure 3 - Supp 1) as well as updated our Methods and Results with our findings. Please refer to our response to Reviewer #2 - Comment 4 to see our revised Methods, Results, and new Figure 3 - Supplement 1.

In regard to whether any objects tended to have more evidence over others across rooms, we did not find this to be the case. We have not included these results in our revised manuscript, but have included a plot below for your perusal.

We have also included a new supplementary figure (Figure 3 - Supplement 1) that shows the global across-participant average and standard deviation in time taken to speak about a room or object. These analyses (see panels C and D below) did not find any significant differences in time taken to speak

about the objects compared to the global mean, suggesting no bias towards a particular object in either guided or free recalls.

New Figure 3 - Supplement 1

Please find below our new Figure 3 - Supplement 1:

Figure 3 - Supplement 1: Additional behavioral measures. (A–D.) Average time participants spent during recall tasks: (A.) Recalling rooms during guided recalls. B. Recalling rooms during free recalls. (C.) Recalling objects during guided recalls. (D.) Recalling objects during free recalls. For A-D, bar charts describe mean recall time across participants, with error bars indicating standard deviation, and dashed gray lines indicating the global mean. Stars above bar plots indicate a significant difference in time from the global mean, for Bonferroni corrected $p < 0.05$. (E.) Contiguous vs noncontiguous mental traversal. When recalling rooms explored in VR, participants either recalled spatially adjacent (contiguous) rooms or jumped between rooms separated by more than one degree (noncontiguous). (F.) Contiguity in free recall. We measured

the proportion of contiguous room transitions during free recall. Across participants, spatially adjacent rooms were recalled more often than expected by chance ($t(24) = 14.19, p < 0.001$) suggesting an unprompted bias towards contiguous mental traversal.

Comment 4: Clarification of Classifier Training

Comment:

“Reading the caption of Figure 3A, it sounds like the classifier was trained on N-1 participants viewing the object to be classified, while the text makes it sound like it was trained on N-1 participants retrieving the object from memory when in its associated room. Which is it (perception or memory), or am I misinterpreting the analysis? (I do see you have a POCN too but this Figure is specifically labeled ROCN).”

Response:

Thank you for bringing this point of ambiguity to our attention. To clarify, object templates (for both POCN and ROCN) were derived from neural activity evoked when participants viewed videos of the objects (i.e., during object perception). To define the ROCN, we trained the object classifier with the N-1 participants' object templates and then tested on the room-viewing video of the left-out-participant in which they were tasked to recall the name of the object paired to the room in the video. In contrast, to define the POCN, we used the same classifier but tested instead on the object-viewing portion of the left-out-participant. In both cases, the left-out participant was never part of our training data and we therefore avoided double-dipping.

Importantly, acquiring POCN also served as a sanity check to identify brain regions involved in object perception. Because of this, it was no surprise the POCN resulted in primarily visual cortex (and no auditory cortex).

We have now revised the caption of Figure 3A (now Figure 4A) and the relevant section in the Methods and Results sections to more accurately reflect this distinction and clarify this potential point of ambiguity.

Modified Figure 4A (previously Figure 3A) caption:

Retrieved object classifier network (ROCN) methodology and surface maps. (A) During the post-learning room-video object recall task, participants watched a video of a room and verbally recalled the object that was paired to it. In a leave-one-participant-out cross-validation procedure, the characteristic object patterns of the N-1 group – evoked during a separate phase of the study in which participants viewed object videos – were used to train a multinomial logistic classifier. This classifier was then applied to each timepoint on the left-out participant’s room-video object recall data. In the pictured example, the left-out participant, Fernando, is recalling the carrot object that was paired with the hexagon room currently being presented. The object classifier, trained on patterns evoked when other participants viewed the objects, was applied to each timepoint of Fernando’s recall. We then measured the fraction of timepoints during the hexagon-room video that were classified as activating the carrot representation. (B.) For each...

Revised Results (starting on line 207):

In these trials, participants viewed videos of all rooms and verbally recalled the object that had been assigned to each room as it was presented (Fig 4A). We used a leave-one-participant-out cross-validation procedure, whereby we made a neural template for each object (using data from a separate phase of the study in which participants viewed object videos) based on data from N-1 participants, and then we used these templates to classify the (not-visibly-present) objects being recalled during room viewing in the held-out participant (Fig 4A).

Revised Methods 1 (starting on line 776):

These object templates, which were obtained for every participant, were then used in subsequent analyses for training multinomial logistic classifiers. All object classifiers described in this manuscript were trained on these perception-evoked patterns.

Revised Methods 2 (starting on line 828):

Similarly, we identified the networks (Perceived Object Classifier Network; POCN) involved in perception of objects, by classifying objects during post-learning object videos. Importantly, to avoid circularity in our analyses, all object classifiers (whether those made for ROCN or POCN) were trained with N-1 object perception data using a leave-one-participant-out procedure 25 times, where testing occurred on the left-out participant.

Comment 5: Object Reinstatement and Correct/Incorrect Recall Trials

Comment:

"In the paragraph starting on line 751 (Section 5.8.5), did this measure of object evidence only use trials where participants correctly recalled the object (e.g., "teddy bear"), or did it also include false recollections? It might help to have a little text justifying the decision since I can see why one would do it either way, but it does slightly change the interpretation (i.e., looking at correct reinstatement, or looking at any time a participant reports recalling an item even if it wasn't really there)."

Response:

Great point –thank you for bringing this to our attention. For any run of a specific guided or free recall, we collected the total object evidence across all timepoints a participant verbally recalled that object, regardless of whether the object was recalled along with the correctly associated room. We did not condition our object reinstatement measure on recall of the correctly associated room because we were interested in studying how pre-learning room reliability affects object recall *in general* (as opposed to studying how reinstatement of a room representation at recall triggers retrieval of the associated object). We have modified the Methods section you highlighted to clarify this point. Additionally, we have added a new figure (Figure 3) that explicitly describes how participant recalls were transcribed and scored for behavioral accuracy – which we hope will also bring clarity into how object and room recall were temporally segmented and evaluated across both behavioral and neural analyses. This new figure also

addresses Reviewer #1's suggestion to expand our description of behavioral analyses (please refer to Reviewer #1 - Comment 3 for the new figure addition). Finally, regarding false recollections, in which a participant may have recalled an object that was not along the presented path: this was exceptionally rare, occurring in only 6 guided recalls out of 275, across all participants. We did not include these false recollections when calculating accuracy, and we show how we accounted for these exceptional cases in the example recall presented in Figure 3.

Revised Methods (starting on line 911):

We used the same leave-one-out cross validation procedure described previously to predict object identity during guided and free recalls. As described previously, we shifted each recall timeseries (11 guided recalls and 1 free recall) by 4 TRs to approximate the HRF delay, and used the multinomial classifier to predict object classes at every time-point for every participant's recalls. Given that that multiclass classifier was trained on all 23 object classes, we obtained a probability distribution across all 23 classes that described the evidence of each class being reinstated at each timepoint. For any run of a specific guided or free recall, we collected the total object evidence across all timepoints a participant verbally recalled that object, **regardless of whether the associated room was also recalled. We did not condition our object reinstatement measure on recall of the correctly associated room because we were interested in studying how pre-learning room reliability affects object recall *in general* (as opposed to studying how reinstatement of a room representation at recall triggers retrieval of the associated object). We then averaged these timepoints across recall runs (guided and free recalls separately).** For example, if during the first guided recall a participant verbally recalled the object "teddy bear" in two **separate chunks of time** for a total of 16 TRs, we collected the classifier probability for "teddy bear" across those 16 TRs and then similarly for every TR "teddy bear" was recalled in all other guided recalls before averaging to get the total "teddy bear" evidence. For a given participant, we did this for each object combining across all 11 guided recalls and, separately, for the participant's single free recall, yielding 23 mean object probabilities for each type of recall task (guided and free recall) for each searchlight.

We wanted to obtain a single value for each object in each participant (separately for guided and free recall), indicating how well that object was reinstated **during recall**. We did this in two ways: By averaging an object probability across all searchlights that formed part of ROCN or POCN, to obtain an overall ROCN or POCN reinstatement score, respectively, for every object and every participant. We used these as our overall network object reinstatement scores in our subsequent analysis.